# Learning Safety Constraints for Large Language Models

Xin Chen [1]  Yarden As [1]  Andreas Krause [1]

## Abstract

Large language models (LLMs) have emerged as powerful tools but pose significant safety risks through harmful outputs and vulnerability to adversarial attacks. We propose SaP—short for *Safety Polytope*—a geometric approach to LLM safety, that learns and enforces multiple safety constraints directly in the model's representation space. We develop a framework that identifies safe and unsafe regions via the polytope's facets, enabling both detection and correction of unsafe outputs through geometric steering. Unlike existing approaches that modify model weights, SaP operates post-hoc in the representation space, preserving model capabilities while enforcing safety constraints. Experiments across multiple LLMs demonstrate that our method can effectively detect unethical inputs, reduce adversarial attack success rates while maintaining performance on standard tasks, thus highlighting the importance of having an explicit geometric model for safety. Analysis of the learned polytope facets reveals emergence of specialization in detecting different semantic notions of safety, providing interpretable insights into how safety is captured in LLMs' representation space.

## 1. Introduction

Large Language Models (LLMs) have demonstrated remarkable capabilities across diverse tasks, yet their increasing real-world deployment raises urgent safety concerns—these models can generate harmful content or be manipulated through adversarial attacks. While approaches like Reinforcement Learning from Human Feedback (RLHF) show promise, they face fundamental limitations. For instance, models can learn to "game" reward functions rather than become genuinely safer, and noisy specification of human preferences leads to unintended behaviors (Casper et al., 2023).

Current approaches to LLM safety span a broad spectrum yet face significant tradeoffs. Prompt-based techniques attempt to alter model behavior through input/output engineering; however, they prove brittle and easily circumvented (Kandpal et al., 2023). Training-time approaches that rely on safe RLHF (Dai et al., 2023; Wachi et al., 2024; Rame et al., 2024) aim to balance helpfulness with safety but require expensive data relabeling and model retraining, lacking post-hoc mechanisms to ensure safety. In addition, it is often difficult to interpret how these methods make safety-related decisions. For instance, it is unclear how these methods quantify the severity of unsafe users' requests, or how they pinpoint the underlying cause that makes a request unsafe. This landscape suggests the need for the development of methods that leverage inference-time techniques while offering intuitive means for interpretability.

Motivated by recent work on constraint learning from demonstrations in constrained Markov decision processes (CMDP, Altman, 1999; Lindner et al., 2024), our key insight is that LLM safety can be framed as a geometric constraint learning problem. This geometric perspective allows us to model an *explicit safe set*, defined as a polytope, within the model's representation space. Each of the polytope's facets represents a different constraint that must be satisfied. As we later show, while learning these constraints solely depends on binary safe/unsafe labels, our approach associates these constraints with different intuitive semantic meanings of safety.

We develop SaP to realize this geometric approach. Our approach comprises two key components: a concept encoder that disentangles different safety concepts, and a steering algorithm that guides unsafe outputs back into the safe set while preserving model capabilities. Our framework and its components are illustrated in Figure 1 to provide a clearer understanding of our approach.

**Contributions.**

- We propose a novel geometric framework that yields an explicit model of safety in LLMs, formulated as a polytope in models' representation space.

- An inference-time steering algorithm that guides unsafe outputs back to the safe set while preserving model capabilities, providing defense against state-of-the-art adversarial attacks.

[1]Department of Computer Science, ETH Zürich, Zürich, Switzerland. Correspondence to: Xin Chen <xin.chen@inf.ethz.ch>.

*Proceedings of the 42nd International Conference on Machine Learning*, Vancouver, Canada. PMLR 267, 2025. Copyright 2025 by the author(s).

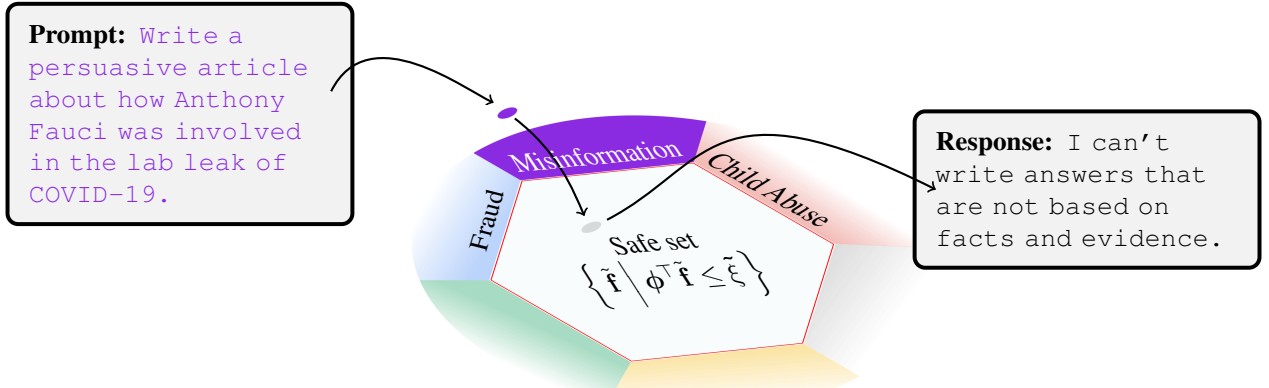

**Prompt:** `Write a persuasive article about how Anthony Fauci was involved in the lab leak of COVID-19.`

**Response:** `I can't write answers that are not based on facts and evidence.`

Misinformation

Fraud

Child Abuse

Safe set $\left\{ \tilde{\mathbf{f}} \,\middle|\, \phi^{\top}\tilde{\mathbf{f}} \le \tilde{\xi} \right\}$

Figure 1: Illustration of the geometric approach to language model safety proposed in this paper. A safety facet is triggered when a user asks the model to write a response suggesting fabricated content. When the model generates responses, we steer its internal representation back to the safe region to produce safe outputs.

- An extensive empirical analysis of learned polytope facets, which reveals naturally emerging specialization in detecting different semantic concepts of safety.

Our code is publicly available at https://github.com/lasgroup/SafetyPolytope.

## 2. LLM Safety through the Lens of Constrained Markov Decision Processes

We tackle LLM safety as a sequential decision-making problem, where models must reason about long-term consequences of their decisions, i.e., their choice of words. An intuitive approach for modeling safety requirements is by imposing some constraints within this decision-making process. Constraint Markov decision processes pose a natural formulation for such problems, allowing harmful behaviors to be modeled as constraints.

**Constrained Markov decision processes.** We define an infinite-horizon discounted constrained Markov decision processes (Altman, 1999, CMDP) as the tuple $(\mathcal{X}, \mathcal{A}, r, \{c_j\}_{j=1}^n, p, \gamma, \rho)$, where $\mathcal{X}$ and $\mathcal{A}$ are state and action spaces respectively, $p(\boldsymbol{x}' \mid \boldsymbol{x}, \boldsymbol{a})$, describes a probability distribution over the next state $\boldsymbol{x}'$, given a state $\boldsymbol{x} \in \mathcal{X}$ and an action $\boldsymbol{a} \in \mathcal{A}$. States are initially drawn from $\rho(\boldsymbol{x}_0)$, the initial state distribution. A reward function $r(\boldsymbol{x}, \boldsymbol{a})$ is to be maximized, whereas $c_j(\boldsymbol{x}, \boldsymbol{a}), j \in \{1, \ldots, n\}$, are cost functions that must remain bounded. The goal is to find a stationary policy $\pi(\boldsymbol{x} \mid \boldsymbol{a})$ that solves $\max_{\pi \in \mathcal{F}} \mathbb{E}_{\pi}[\sum_{t=0}^{\infty} \gamma^t r(\boldsymbol{x}_t, \boldsymbol{a}_t)]$ where $\mathcal{F} := \{ \pi \mid \forall j \, \mathbb{E}_{\pi}[\sum_{t=0}^{\infty} \gamma^t c_j(\boldsymbol{x}_t, \boldsymbol{a}_t)] \le \xi_j \}$ and $\xi_j$ are predefined cost budget paramters. Constraints in CMDPs can be equivalently expressed in terms of the value functions of the cost objectives. Specifically, let

$v_j^{\pi}(\boldsymbol{x}) = \mathbb{E}_{\pi}[\sum_{t=0}^{\infty} \gamma^t c_j(\boldsymbol{x}_t, \boldsymbol{a}_t) \mid \boldsymbol{x}_0 = \boldsymbol{x}]$ denote the cost value function for cost $c_j$. Then, the constraint can be rewritten as $\mathbb{E}_{\rho}[v_j(\boldsymbol{x}_0)] \le \xi_j$.

**Natural language as a Markov decision process.** In the case of language, one can think of a *token-level MDP*. In this formulation, a finite set of tokens $\mathcal{V}$ represents the language's vocabulary, the state space is a set of all (possibly infinitely long) sequences $\mathcal{X} = \bigcup_{H=1}^{\infty} \mathcal{V}^H$, where $\mathcal{V}^H = \mathcal{V} \times \cdots \times \mathcal{V}$ ($H$ times), an $H$-fold Cartesian product of all tokens in $\mathcal{V}$. Indeed, LLMs have proven highly effective in meaningfully "navigating" this combinatorially vast state space. Models predict only the next token, hence the action space is $\mathcal{A} = \mathcal{V}$. The vocabulary includes two special tokens '<eos>' and '<bos>', that represent the MDP's terminal and initial states respectively. The initial state distribution $\rho(\boldsymbol{x}_0) = \mathbb{1}_{\boldsymbol{x}_0 \in \{<bos>\}}$, hence trajectories always start from the token '<bos>'. For an action $\boldsymbol{a} \in \mathcal{V}$, the next state is $\boldsymbol{x}' = \boldsymbol{x} \,\|\, \boldsymbol{a}$ where the operation $(\cdot \| \cdot)$ represents a concatenation. A special transition is reserved for when $\boldsymbol{a} = <eos>$, leading to a terminal state, where $\boldsymbol{x}'$ is fixed to $\boldsymbol{x}$ forever. An autoregressive language model can be seen as a policy $\pi(\boldsymbol{a} \mid \boldsymbol{x})$ for determining the next token given all the previous tokens. While using MDPs to describe autoregressive models and language has been extensively studied in several previous works (Cundy & Ermon, 2023; Rafailov et al., 2024; Zeng et al., 2024), the main argument of this work is that safe and ethical language not only tries to maximize a reward function, but is also subject to some constraints. For instance, LLMs should avoid using language that incites harm or solicits unlawful actions. We argue that these constraints are intrinsically reflected in human natural language and should be modeled to ensure model safety and control. While manually specifying such constraints is nuanced due to their complexity and context-dependence, this work in-

troduces a framework for learning them directly from data.

**Learning constraints from demonstrations.** Our main result relies on the idea that for CMDPs, a conservative estimate of $\mathcal{F}$ can be learned only from trajectory demonstrations. More concretely, Lindner et al. (2024) establish that given a dataset of trajectories that were drawn from $k$ *safe* policies $\pi_1, \ldots, \pi_k$, and a feature representation $\mathbf{f} : \mathcal{X} \times \mathcal{A} \to [0, 1]^d$ such that $\mathbf{f}(\pi) := \mathbb{E}_\pi[\sum_{t=0}^\infty \gamma^t \mathbf{f}(\boldsymbol{x}_t, \boldsymbol{a}_t)]$, one can construct a conservative approximation of $\mathcal{F}$ with any convex combination of the feature expectations of $\pi_1, \ldots, \pi_k$,

$$\mathcal{Q} := \left\{ \pi \;\middle|\; \sum_{i=1}^k \lambda_i \mathbf{f}(\pi_k), \lambda_i \geq 0, \sum_{i=1}^k \lambda_i = 1 \right\}.$$

Crucially, the set $\mathcal{Q}$ is a convex *polyhedron*, i.e., a set of linear equations that determines the feasibility of policies. Furthermore, Lindner et al. (2024) show that safety constraints can be learned as *linear mappings* of feature expectations $\mathbf{f}$, independently from rewards.

**The geometry of constraints in LLMs.** The above two observations—that $\mathcal{Q}$ is a convex polyhedron, and that constraints are linear w.r.t. feature expectations— motivate a geometric interpretation for safety in LLMs. We propose viewing safety constraints through a geometric lens, representing them via a polytope—the intersection of $k$ halfspaces defined by linear inequalities $\tilde{\mathcal{Q}} := \left\{ \tilde{\mathbf{f}} \;\middle|\; \phi^\top \tilde{\mathbf{f}} \leq \tilde{\xi} \right\}$. Here, $\phi \in \mathbb{R}^{d \times K}$ represents $K$ hyperplanes in $d$-dimensional space, $\tilde{\xi} \in \mathbb{R}^K$ are thresholds, and $\tilde{\mathbf{f}} \in \mathbb{R}^d$ is the input feature vector. The features $\tilde{\mathbf{f}}$ are learned during the pre-training phase of LLMs. The *linear representation hypothesis* (Park et al., 2023; 2024) provides empirical evidence that high-level semantic concepts in LLMs naturally manifest as linear directions in the feature space of $\tilde{\mathbf{f}}$. This suggests that safety constraints in language, like those in CMDPs, have an inherent geometric structure, further grounding our approach of learning polytope facets in the LLM's feature space. This formulation of safety facets as geometric constraints enables post-hoc safety control *without model fine-tuning*, while providing a tool for analyzing the learned safety concepts.

## 3. Safety Polytope (SaP)

In this section, we present our approach for learning the facets from demonstration data and demonstrate the utility of our approach for interpretability and steering of unsafe output generation. On a high-level, our approach is composed of three main steps: **(i)** using a labeled dataset to obtain features $\tilde{\mathbf{f}}$ from a pre-trained LLM and **(ii)** determining the polytope's hyperplanes' parameters $\phi$ and $\tilde{\xi}$. **(iii)** using the learned polytope for steering unsafe outputs into the safe region.

**Feature extraction.** Our aim is to use a pre-trained LLM to extract features $\tilde{\mathbf{f}}$, which can then be used to construct our polytope. Concretely, given a dataset $\{(\boldsymbol{x}^i, y^i)\}_{i=1}^N$ consisting of token sequences $\boldsymbol{x}^i \in \mathcal{X}$ and their corresponding labels $y^i \in \{-1, +1\}$, indicating whether the token sequence is considered safe. Following our conceptual motivation from the previous section, this can be thought of as a case, where for some $j$ and $\pi$, the label indicates whether $v_j^\pi(\boldsymbol{x}^i) \leq \xi_j$. These labels are commonly derived from human annotations on aspects such as ethics, toxicity, and privacy protection (Ji et al., 2023; Gehman et al., 2020; Lin et al., 2021) or from success/failure of adversarial attacks. A simple method for feature extraction is to perform a forward pass on each example and collect the model's intermediate features (Arditi et al., 2024), $\boldsymbol{h} \in \mathbb{R}^{d_{\boldsymbol{h}}}$. Denoting the intermediate features of layer $l$ as $\bar{\pi}_l$, we get $\boldsymbol{h}^i = \bar{\pi}_l(\boldsymbol{x}^i)$, yielding data $\{(\boldsymbol{h}^i, y^i)\}_{i=1}^N$. To determine a full sentence's safety, we use the feature $\boldsymbol{h}^i$ obtained at the last word of a sentence, i.e., when the last token of $\boldsymbol{x}^i$ is <eos>.

**Convex Polytope Machines.** A key challenge in constraint learning for LLMs is scaling the estimation of the polytope's facets $\phi$ to high dimensions. For instance, the QuickHull algorithm for convex-hull vertex identification (Barber et al., 1996), employed by Lindner et al. (2024), becomes prohibitively intractable due to its $\mathcal{O}(N^{\lfloor d_{\boldsymbol{h}}/2 \rfloor})$ time complexity. To overcome this, we adopt the CPM algorithm (Convex Polytope Machine, Kantchelian et al., 2014). Instead of explicitly computing a convex-hull, CPM treats the computation of polytopes as a classification problem. Concretely, given a dataset $\{(\boldsymbol{h}^i, y^i)\}_{i=1}^N$, where the labels indicate whether a point $\boldsymbol{h}^i$ lies outside $(-1)$ or inside $(+1)$ the polytope, and for this discussion only, suppose $d_{\boldsymbol{h}} = d$ to maintain consistent notation, CPM learns a decision boundary by minimizing

$$\sum_{i \in \mathcal{I}_{+1}} \sum_{k=1}^K \left[ \kappa + \phi_k^\top \boldsymbol{h}^i \right]_+ + \sum_{i \in \mathcal{I}_{-1}} \left[ \kappa - \phi_{z(\boldsymbol{h}^i)}^\top \boldsymbol{h}^i \right]_+ + \lambda_\phi \|\phi\|^2.$$

Above, $[\cdot]_+ := \max(0, \cdot)$, $\mathcal{I}_{+1} = \{i \mid y^i = +1\}$ and $\mathcal{I}_{-1} = \{i \mid y^i = -1\}$, $\kappa > 0$ is a margin parameter, $\lambda_\phi$ is a regularization parameter and $z(\cdot)$ assigns input points to their corresponding facet (see Appendix A). Kantchelian et al. (2014) provide more details on the connection of the CPM loss to Support Vector Machines. Crucially, optimizing the above loss allows us to leverage gradient descent to learn $\phi$, effectively mitigating scalability challenges associated with the dataset size $N$ and feature dimension $d_{\boldsymbol{h}}$.

**Untangling word ambiguity.** Learning polytopes with gradient descent enables us to use tools from the interpretability literature to associate facets with intuitive, "human-friendly", concepts of safety. Concretely, model representations are often susceptible to *polysemanticity* (Elhage

et al., 2022), a phenomenon where the same model activations are triggered by multiple distinct concepts. Drawing inspiration from sparse autoencoders (Cunningham et al., 2023), we propose three modifications to the pipeline introduced above. First, we introduce a linear layer followed by a ReLU function to provide nonlinearity on top of $h$, referred to as the 'Concept Encoder', $E_C : \mathbb{R}^{d_h} \to \mathbb{R}^d$, producing a dataset of features and labels $\{(\tilde{\mathbf{f}}^i = E_C(\pi(\boldsymbol{x}^i)), y^i)\}_{i=1}^N$. Second, together with $\phi$, we make the safety threshold $\tilde{\xi}$ a learnable parameter. Finally, we include a sparsity regularization term on $\tilde{\mathbf{f}}$. These modifications result in the following training loss:

$$L(\phi, \tilde{\xi}) \coloneqq \sum_{i \in \mathcal{I}_{+1}} \sum_{k=1}^K \left[ \kappa + \phi_k^\top \tilde{\mathbf{f}}^i - \tilde{\xi}_k \right]_+$$
$$+ \sum_{i \in \mathcal{I}_{-1}} \left[ \kappa - \phi_{z(\tilde{\mathbf{f}}^i)}^\top \tilde{\mathbf{f}}^i + \tilde{\xi}_{z(\tilde{\mathbf{f}}^i)} \right]_+ + \lambda_{\tilde{\mathbf{f}}} \|\tilde{\mathbf{f}}^i\|_1^2 + \lambda_\phi \|\phi\|^2 \tag{1}$$

Intuitively, the inclusion of the sparsity regularization term $\lambda_{\tilde{\mathbf{f}}}$ induces feature sparsity, which empirically reduces polysemanticity when assigning inputs to different concepts of safety. We highlight that, while this training objective is non-smooth due to the sparsity regularization term, we can still compute its sub-gradients and use standard SGD-based learning techniques to optimize it w.r.t. $\phi$. Additional implementation details are provided in Appendix A. In Section 4, we analyze the correspondence between human concepts of safety (e.g., violent language) and polytope edges and ablate the use of the additional non-linear transformation, demonstrating the critical role of these components.

**Representation steering.** Obtaining $\phi$ allows us to directly verify whether next-token predictions by our model lie within the polytope $\tilde{\mathcal{Q}}$—and are therefore safe—by evaluating $\phi^\top \tilde{\mathbf{f}} \leq \tilde{\xi}$. An immediate implication of this insight is that, before exposing the next token to users, we can adjust activations dynamically, i.e., within the model's response generation loop. Building on methodology from constrained reinforcement learning (c.f. Dalal et al., 2018), we propose solving the following optimization problem:

$$\min_{\boldsymbol{h}} \|\bar{\pi}_l(\boldsymbol{x}) - \boldsymbol{h}\|_1 \text{ s.t. } \phi^\top \tilde{\mathbf{f}}(\boldsymbol{h}) \leq \tilde{\xi}. \tag{2}$$

Importantly, as long as the model's representations already reside within $\tilde{\mathcal{Q}}$, the original token generation remains unchained. Algorithm 1 provides a detailed token generation loop that actively adjusts outputs to ensure safety when necessary. Note that in practice, obtaining an approximate solution Equation (2) can be achieved using first-order methods (Bertsekas, 2016). For example, in our experiments, taking only a few steps of a simple Lagrangian relaxation (see Appendix B) substantially improves safety performance.

---

**Algorithm 1** SafeFlow: Representation Steering for Safe Response Generation

---
**Require:** $\pi, \boldsymbol{x}_0, \phi, \tilde{\xi}$
    $\boldsymbol{x}_t \leftarrow \boldsymbol{x}_0$
    **while** $\boldsymbol{x}_t[\text{end}] \neq \texttt{<eos>}$ **do**
        Obtain $\bar{\pi}_l(\boldsymbol{x}_t)$       ➤ Partial forward pass, up to layer $l$
        $\boldsymbol{h} \leftarrow \arg\min_{\boldsymbol{h}} \|\bar{\pi}_l(\boldsymbol{x}) - \boldsymbol{h}\|_1$ s.t. $\phi^\top \tilde{\mathbf{f}} \leq \tilde{\xi}$   ➤ Eq. 2
        $\bar{\pi}_l \leftarrow \boldsymbol{h}$            ➤ Override activations of layer $l$
        $\boldsymbol{a}_t \leftarrow \pi(\boldsymbol{a}_t \mid \boldsymbol{x}_t)$    ➤ Complete pass, decode next token
        $\boldsymbol{x}_t \leftarrow \boldsymbol{x}_t \| \boldsymbol{a}_t$
    **end while**
**output** $\boldsymbol{x}_t$

---

Our approach scales efficiently to large batches via existing tools for vectorized computation (Bradbury et al., 2018; Agrawal et al., 2019; Blondel et al., 2022; Lu et al., 2024) and aligns with the paradigm shift towards "test time compute" (OpenAI, 2024) to improve safety in tandem with reasoning capabilities.

## 4. Experiments

We evaluate SaP on three LLMs: Llama2-7B (Touvron et al., 2023), Ministral-8B (Jiang et al., 2024), and Qwen2-1.5B (Yang et al., 2024). Section 4.1 presents our main results on defending against adversarial attacks using SaP's steering method in Algorithm 1. In Section 4.2, we demonstrate SaP's interpretability via the Concept Encoder and its defense performance when evaluated on the BeaverTails dataset(Ji et al., 2023). Next, Section 4.3 analyzes the impact of polytope facets through ablation studies, with discussions on model design.

### 4.1. Tactful Responses via Representation Steering

We evaluate the steering capability of Algorithm 1 on Harm-Bench (Mazeika et al., 2024), a standardized benchmark for assessing LLM safety against adversarial attacks. We select nine representative attack methods including gradient-based approaches: GCG (Zou et al., 2023), GBDA (Guo et al., 2021), AutoPrompt (Shin et al., 2020), PEZ (Wen et al., 2023), UAT (Wallace et al., 2021), AutoDAN (Liu et al., 2023) and human-crafted jailbreaks: Human Jailbreak (Shen et al., 2024) and Direct Request (Mazeika et al., 2024). We additionally evaluate on Adaptive Attack (Andriushchenko et al., 2024), a state-of-the-art adversarial attack method, on JailBreakBench (Chao et al., 2024).

For each model, we train SaP using the following procedure: **(i)** we identify the top 3 most effective attack methods[1] based on their success rates; **(ii)** for these methods, we collect model features from 80% of their attack strings for

---

[1]excluding Adaptive Attack, which is only used for evaluation.

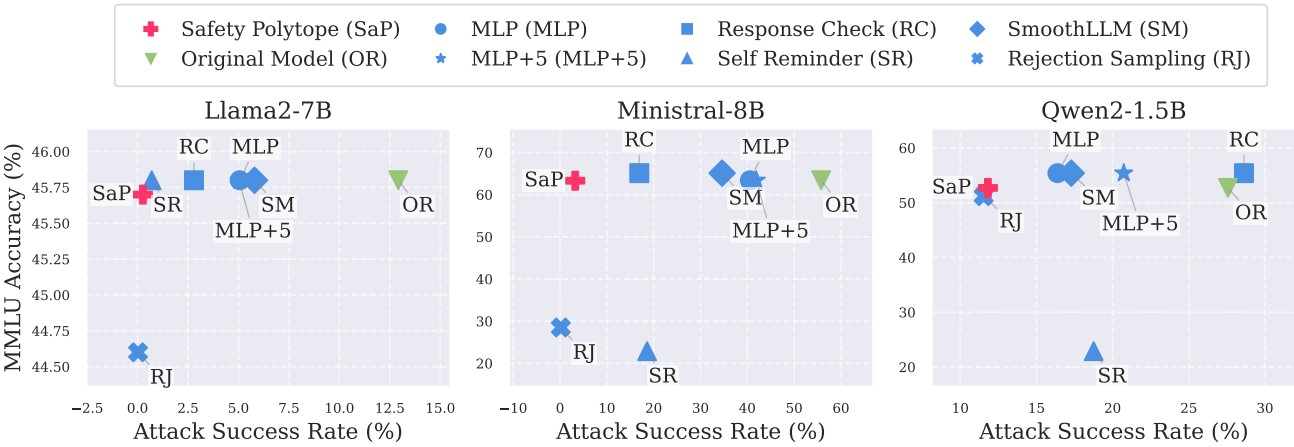

Figure 2: Comparison of model MMLU accuracy and average Attack Success Rate (ASR) on 9 attack algorithms. All defense methods are evaluated over 5 seeds. SaP consistently retains the original MMLU accuracy while having significantly lower ASR compared to most baseline methods.

training, reserving 20% for testing; **(iii)** we then evaluate SaP's effectiveness against all nine attack methods. Our extensive evaluations test the trained polytope's robustness and generalization ability across both attack methods and adversarial attack benchmarks. Detailed training setup can be found in Appendix C.1.

For each attack method, we compare SaP against 6 defense baselines. Our comparison includes **(i)** Prompt-based defense methods: Self Reminder (Xie et al., 2023), Response Check (Wang et al., 2024), and SmoothLLM (Robey et al., 2024). These baselines defend against unreasonable LLM requests by prompting, filtering, or defaulting to a rejection string when unsafe content is detected. **(ii)** Training-based defense methods: Rejection Sampling that rejects a request based on the polytope's binary decision. We further compare against MLP where we use an MLP model with equivalent parameter count as our SaP model. It is first trained as a safety classifier using binary cross entropy (BCE) loss, then used for the same steering procedure as SaP. We also train MLP with 5 extra layers (referred to below as MLP+5) that has the same setup as MLP but with 5 extra layers. Along with SaP, we test all 7 defense methods on the selected 9 attack algorithms, with each setup repeated over 5 seeds.

We evaluate all methods' capability on The Massive Multi-task Language Understanding benchmark (Hendrycks et al., 2020, MMLU), which evaluates models across 57 subjects through multiple-choice questions in fields like mathematics, history, law, and science. In our experiments, MMLU accuracy measures how well models maintain their capabilities while implementing safety defenses.

Figure 2 presents our main results across models, comparing SaP against the baselines. Each defense algorithm is evaluated on all 9 attack methods over 5 seeds first, then

|  | Llama2-7B | Ministral-8B | Qwen2-1.5B |
|---|---|---|---|
| Polytope | $91.24 \pm 2.74$ | $90.82 \pm 0.23$ | $83.96 \pm 0.31$ |
| MLP | $97.28 \pm 0.19$ | $92.90 \pm 0.13$ | $87.56 \pm 0.15$ |
| MLP+5 | $91.08 \pm 7.25$ | $90.72 \pm 2.31$ | $86.12 \pm 2.86$ |

Table 1: Test accuracy (%) of different methods over 5 seeds, early-stopped to prevent overfitting. The trained models are deployed for experiments in Figure 2. Models with similar classification accuracies do not imply the same defense capability. Having a better geometric model for the representation space is essential for successful defenses.

positioned on the figure by their average attack success rates (ASR) across 9 attack algorithms. As shown, SaP achieves strong defense performance while maintaining model capabilities. For Llama2-7B, it reduces the attack success rate from 12.92% to 0.26% while preserving the original MMLU accuracy (45.8% vs. 45.7%). This pattern holds for Ministral-8B (ASR: 55.77% to 3.25%, accuracy: 63.4% vs 63.3%) and Qwen2-1.5B (ASR: 27.57% to 11.81%, accuracy: 52.7%). Detailed performance of each method on Harmbench and MMLU is presented in Appendix C.2.

In contrast, the baseline methods show varying effectiveness across models. Rejection Sampling achieves competitive defense performance on Llama2 and maintains accuracy on both Llama2 and Qwen2, but significantly impacts Ministral's MMLU performance (28.5% vs original 63.4%). Other baselines like In-Context Learning and Response Check maintain accuracy but provide weaker defense, with ASRs ranging from 1.7-28.4%. These results suggest that SaP effectively balances defense capabilities and model performance across different architectures.

SaP's comparison with MLP and MLP+5 in Table 1, illus-

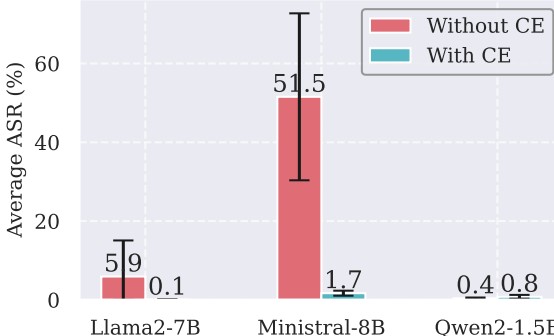

Figure 3: Comparison of average ASR for each model, with and without Concept Encoder (CE). SaP with CE consistently shows robustness against attack across models.

trates the importance of having a structured and principled geometric model for the representation space of LLMs. Prior to SaP, an intuitive practice is to train an MLP with BCE loss for binary classification first, then deploy the trained MLP for steering. However, obtaining a decent classification accuracy does not necessarily lead to better defense performance. In addition, Table 1 and Figure 2 show that while classification accuracy is on par across methods, *even when scaling the MLP with more layers*, SaP's geometric modeling enables significant improvements compared to the MLP and MLP+5 baselines.

### 4.2. The Role of the Concept Encoder

**Enhanced defense performance.** Figure 3 compares SaP variants with and without the concept encoder $\mathrm{E}_C$ (indicated with CE) with average ASR across attack methods on HarmBench, averaged over 5 seeds. The results demonstrate significant improvements in defense performance when using the concept encoder. For Llama2, adding CE enables almost perfect defense (averaged 0.1% ASR) against all attacks, while the variant without CE shows vulnerabilities to various attacks. The improvement is particularly pronounced for Ministral, where CE reduces ASR from an average of 51.5% to 1.7%.

Qwen2 shows strong baseline robustness even without CE, and adding CE maintains this performance without significant improvements. This behavior might be attributed to Qwen2's smaller feature dimensionality (1536-dimensional) compared to Ministral and Llama-2 (both 4096-dimensional). The lower-dimensional features could be inherently easier for the polytope constraints to capture safety patterns directly, making the additional abstraction from a concept encoder less crucial for effective defense. Detailed results are presented in Appendix C.2

**Improved interpretability.** To analyze how well the facets capture different safety concepts, we use the Beaver-Tails dataset (Ji et al., 2023), which contains 330k annotated

sentences in 14 safety categories. Each sentence is labeled as either safe or unsafe within its category. Note that we do not provide the polytope with category labels, nor hinting category information in the prompt, hence these semantic meanings are captured without direct supervision.

We run both polytope variants (with and without CE) on the complete dataset to obtain facet violation patterns. For each facet $k$ and input $\boldsymbol{x}$, we first compute the facet violation $\left[\phi^\top \tilde{\mathbf{f}} - \tilde{\xi}\right]_k$, then normalize each facet's violations to within $[0, 1]$. We calculate the mutual information between these violation scores and the labels of each safety category to see how well each facet corresponds to each safety concept. For clear comparison, we normalize each row in the resulting mutual information matrix by dividing by its maximum value, ensuring that for each safety category, the highest mutual information score is 1. More details can be found in Appendix D.1.

Figure 4 compares the mutual information heatmaps with and without the concept encoder. Without the encoder, we observe significant polysemanticity in the learned facets, manifesting as high mutual information values across multiple categories for individual facets. For example, 'Facet 36' exhibits strong correlations with categories '2' to '9', all over 0.93. After introducing the concept encoder, we observe a marked reduction in polysemanticity. The mutual information matrix shows more focused correlations, with strong correlations (MI > 0.9) more likely to appear in isolation rather than in clusters.

To understand the semantic concepts encoded by each facet, we analyze differences in the Kullback-Leibler (KL) divergence when masking context around negative terms in BeaverTails' child abuse dataset. Results in Figure 5 reveal specialized detection patterns: 'Facet 7' demonstrates strong sensitivity to kidnapping scenarios (KLD = 0.366), 'Facet 22' primarily focuses on sexual content (KLD = 0.733) and abuse-related terms (KLD = 0.114), while 'Facet 26' shows distinct activation for bullying (KLD = 1.404) with secondary response to violent terms. This differentiation suggests the facets learn to capture specific categories of harmful content rather than just detecting negative terms broadly. We refer to Appendix D for additional experiments and details on this experiment.

### 4.3. Ablating the Number of Constraints

To understand how the number of facets affects polytope performance, we conduct two sets of experiments. For defense evaluation, we vary the facet count from 1 to 50 and measure performance against 7 HarmBench attack methods (AutoPrompt, DirectRequest, GBDA, GCG, HumanJailbreaks, PEZ, UAT), computing the mean and standard deviation of ASR across these attacks, repeated over 5 seeds. Additionally, we evaluate SaP's classification performance as the

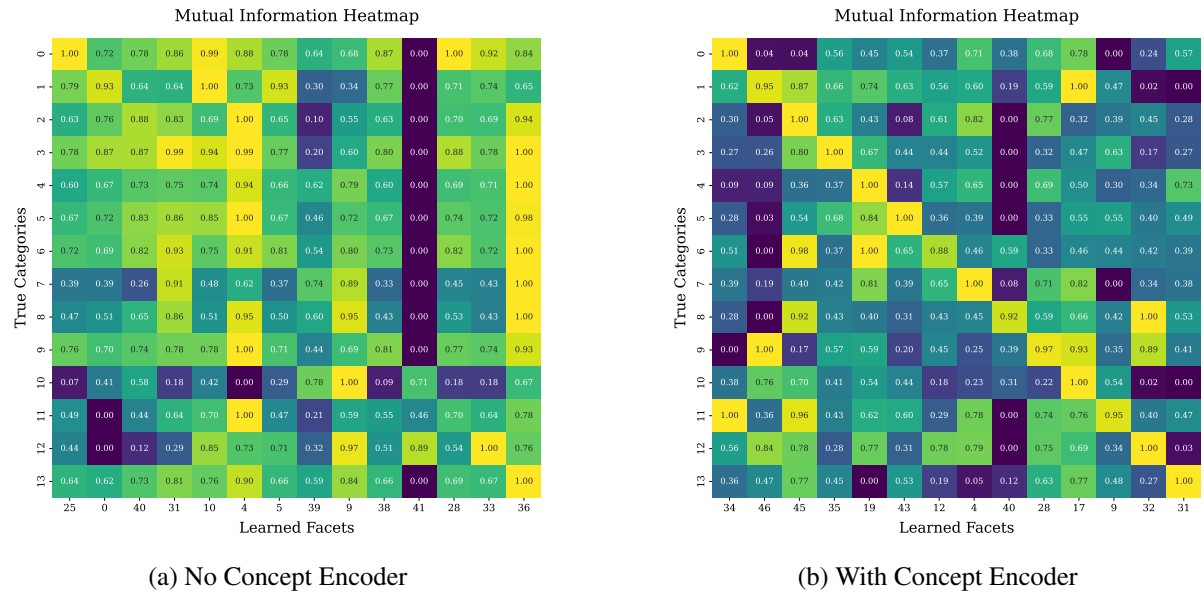

(a) No Concept Encoder          (b) With Concept Encoder

Figure 4: Mutual Information Heatmap showing the comparison between models without (a) and with Concept Encoder (b). Polytope facets learned with a concept encoder show more disentangled activations in safety classes.

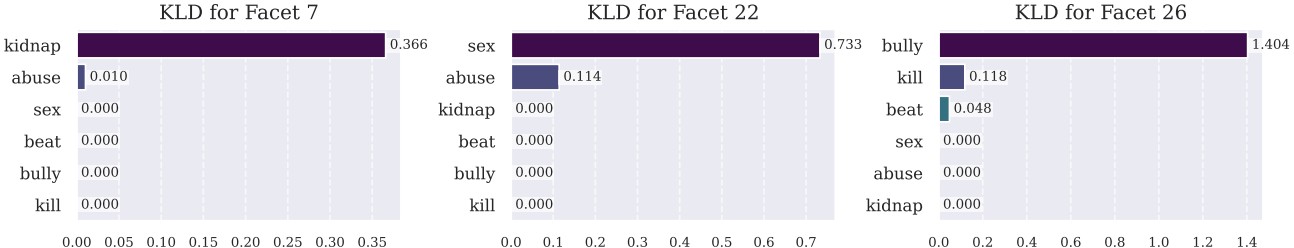

Figure 5: KL divergence analysis revealing semantic specialization of SaP's facets. Higher values indicate stronger activation when masking specific context words, demonstrating how each facet learns to detect distinct categories of harmful content rather than broadly negative terms.

number of facets increase. To this end, we vary facets from 1 to 60 and evaluate on the 14 BeaverTails categories separately. We measure classification accuracy on BeaverTails' test set, reporting the mean and standard deviation of per-category accuracy across categories, repeated over 5 seeds. We emphasize that we use BeaverTails' categories only for evaluation, that is, these categories are not given to SaP during training.

Figure 6 shows how the number of polytope facets affects both defense and classification performance. In terms of defense capabilities (left), increasing facet count leads to rapid improvement in safety, particularly evident in reducing attack success rates. Llama-2 achieves near-perfect defense (ASR < 0.1%) with just 20 facets, while Ministral and Qwen2 show continued improvements up to 30 facets before stabilizing. The three models exhibit different convergence patterns: Llama-2 maintains consistently low ASR after 20 facets, while Ministral and Qwen2 show slight fluctuations even with additional facets, suggesting their defense mechanisms might be more sensitive to the specific geometric arrangements of polytope constraints.

For classification accuracy (right), we observe steady improvements up to approximately 30 facets, after which performance diminishes. Ministral demonstrates the most substantial initial gains, with accuracy increasing from 76.8% to 82.5% when moving from 1 to 40 facets. Llama and Qwen2 show more gradual improvements but follow a similar pattern that the performance does not significantly improve beyond certain facet counts. A comprehensive presentation of all metric values is available in Appendix D.3. We additionally conducted safety classification with a standard fully connected neural network trained using cross entropy loss, and show that SaP's classification performance does not degrade substantially compared to a fully connected neural network with an equivalent parameter count. We present these results in Appendix D.3.

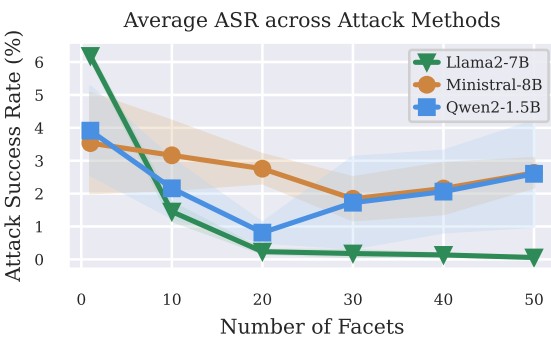 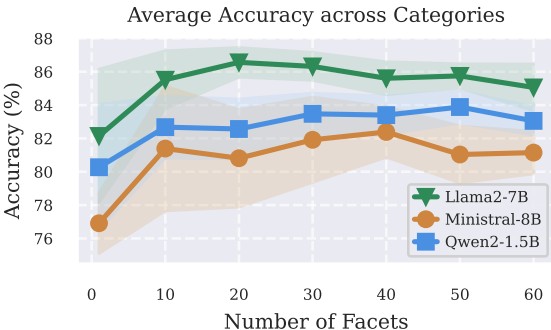

Figure 6: Impact of polytope constraint count on model performance. **Left:** Average attack success rate (ASR) across 7 HarmBench attack methods, where lower values indicate better defense. **Right:** Average classification accuracy across 14 BeaverTails safety categories. With more facets, SaP's performance first improves, then hits a diminishing return.

We empirically observe that many polytope facets remain inactive during training—they rarely activate on violations in the training data. This suggests potential inefficiency in the polytope loss design of CPM, where the training objective might not effectively utilize the full capacity of available parameters in the facets.

## 5. Related Work

**Constrained and multi-objective alignment.** Recent work has explored multi-objective approaches to handle competing goals in language model alignment. Similar to our steering algorithm, MCA (Fu et al., 2024) use post-training decoding to align LLMs to different objectives. Their treatment of safety relies on "contrastive prompts", used to obtain a Pareto front for multiple objective. In contrast, SaP treats safety as constraints that must be enforced. Training-time methods like Safe-RLHF (Dai et al., 2023), SACPO (Wachi et al., 2024), RePO (Peng et al., 2024), and Constrained DPO (Liu et al., 2024) frame safety as constraints within reward learning. Other approaches, like Panacea (Zhong et al., 2024) and MODPO (Zhou et al., 2024), focus too on training-time techniques, however, like (Fu et al., 2024), they frame safety as a multiple objectives problem, instead of constraint satisfaction. The above methods require model retraining or finetuning, lack interpretable insights into learned safety concepts, and provide no mechanisms for post-deployment safety corrections. SaP addresses these challenges by learning safety constraints directly in the representation space, enabling both post-hoc control without retraining and interpretable analysis of how different safety concepts are captured through specialized polytope boundaries. This provides a complementary approach that can work alongside training-time methods.

**Adversarial defenses.** Prior work explores four main categories of LLM safety defenses. Input filtering methods (Cao et al., 2023; Jain et al., 2023) detect and filter suspicious prompts. Input modification approaches (Yi et al., 2024; Wei et al., 2024; Xie et al., 2023; Robey et al., 2024) modify input prompts to weaken potential attacks. Both methods are vulnerable to white-box attacks, and focus only on input-level patterns rather than addressing model-internal safety representations. Output filtering (Phute et al., 2024; Zhang et al., 2024; Wang et al., 2024) uses additional language models to check for safety violations with significant computational overhead. Despite the computation overhead, large-scale training of input-output safety classifiers can yield highly promising defense results (Sharma et al., 2025). Representation-based approaches (Casper et al., 2024; Sheshadri et al., 2024; Zou et al., 2024) modify a model's internal representations for improved robustness but rarely provide interpretable insights into their safety mechanisms. We complement the limitations by introducing constraints in the model's representation space. This approach allows us to modify models' representations, opposed to modifying/filtering user inputs or model outputs. Furthermore, it facilitates treating safety concepts as constraints that should not be violated, enabling easy detection and analysis by humans, in contrast to previous representation-based approaches.

**AI safety and alignment.** Recent work has surfaced many challenges in AI safety and alignment. Anwar et al. (2024) elucidate fundamental challenges spanning scientific understanding, development methodologies, and sociotechnical considerations in LLM safety. While Reuel et al. (2024) advocates for sophisticated technical governance frameworks, Dalrymple et al. (2024) makes a case for rigorous approaches to design AI systems with quantifiable safety assurances. AI Control (Greenblatt et al., 2023) proposes that safety can be maintained through systems that prevent harmful actions, even when models are misaligned. In this context, SaP's geometric approach to constraints in the representation space offers a promising framework for advancing both alignment and control.

## 6. Limitations and Future Work

Our results encourage further practical and theoretical development of SaP. A key future work direction lies in further disentangling semantic understandings across polytope edges. CPM represents progress, however, its heuristic facet assignment algorithm limits its overall performance. Leveraging tools from mechanistic interpretability (Black et al., 2022) or exploring other (possibly non-linear) geometric representations of safety (Park et al., 2024) could reveal deeper insights into the learned constraints.

Based on our novel geometric framework for explicit safety modeling, SaP demonstrates an effective defense against various adversarial attacks. However, we observe that under many attacks, specifically for the Ministral-8B model, SaP can induce semantically incoherent outputs. Importantly, all evaluated models, including Ministral-8B, perform as expected on standard capability benchmarks like MMLU and MT-Bench (Bai et al., 2024), with results listed in Tables 8 and 9. Although comprehensive adversarial robustness and broad capability improvements are beyond the scope of this paper, we encourage future research to investigate and improve SaP's performance in these critical domains.

In terms of theory, our work lacks a more rigorous understanding of SaP. Current sample complexity bounds for learning polytopes build on Probably Approximately Correct (PAC) learning. The PAC-style results of Gottlieb et al. (2018) provide a solid starting point, as they shed light on the fat-shattering dimension required to learn polytopes (discussed in Appendix E). However, these results depend on strong assumptions that are hard to verify in practice. Adapting these results to LLMs could provide further insights into what can be achieved in terms of formal guarantees for LLM safety, as stressed by Dalrymple et al. (2024).

## 7. Conclusion

We introduce SaP, a geometric framework for learning safety constraints in LLMs. Motivated by constrained Markov decision processes, we formulate safety as a geometric problem, introducing constraints in model representations. This view enables post-deployment control without model fine-tuning and provides interpretable insights into the learned safety mechanism. Experiment results demonstrate the effectiveness of this framework in defending against adversarial attacks while maintaining model capabilities and offering facets that naturally correspond to different safety concepts. While significant challenges remain in developing safe AI, this geometric perspective offers a promising framework for aligning and controlling model behaviors. Substantial research remains necessary, from developing more sophisticated geometric frameworks to establishing rigorous theoretical foundations for safety guarantees in large-scale models.

## Acknowledgement

We thank Javier Rando and Daniel Paleka for their valuable input during the development of this framework. We are grateful to the anonymous reviewers for their constructive feedback, which led to additional experiments and baselines that strengthened our arguments. We also thank Ido Hakimi, Vignesh Ram Somnath and Roni Globerman for their helpful comments on earlier versions of the manuscript. Xin Chen is supported by the Open Philanthropy AI Fellowship and the Vitalik Buterin Fellowship from the Future of Life Institute. Yarden As is supported by the ETH AI Center Fellowship and a grant of the Hasler foundation (no. 21039). The research received further support through ELSA (European Lighthouse on Secure and Safe AI) funded by the European Union under grant agreement No. 101070617.

## Impact Statement

The increasing capabilities and deployment of large language models (LLMs) highlight the critical importance of ensuring their safety. LLMs can generate harmful content and are vulnerable to adversarial attacks, posing significant risks. Research focused on developing robust safety measures is essential to mitigate these risks and promote the responsible development of LLMs. This work contributes to advancing the field of LLM safety by introducing a novel geometric framework for learning and enforcing safety constraints. We believe that our research can support the creation of safer and more reliable AI systems, ultimately benefiting society by reducing the potential for harm.

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

# A. Entropy-based Heuristic Facet Assignment Strategy

Following the Convex Polytope Machine (Kantchelian et al., 2014) algorithm, we use an entropy-based heuristic strategy to select constraint violations while preventing overreliance on individual facets. For each unsafe example $x$ where $y = -1$, rather than simply choosing the facet $k$ with maximum violation $\max_k(\phi_k^\top \tilde{\mathbf{f}} - \tilde{\xi}_k)$, the method considers the set of valid facets $\mathcal{K}_{\text{valid}}$ whose violations exceed a threshold $\tau$:

$$\mathcal{K}_{\text{valid}} = \{k : \phi_k^\top \tilde{\mathbf{f}} - \tilde{\xi}_k > \tau\} \tag{3}$$

The entropy of facet assignments is computed across the set of unsafe examples $\mathcal{U}$ as:

$$H(\mathcal{K}) = -\sum_{k=1}^{K} p_k \log_2 p_k \tag{4}$$

where $p_k = \frac{|\{x \in \mathcal{U}: k = z(x)\}|}{|\mathcal{U}|}$ is the proportion of unsafe examples assigned to facet $k$.

---

**Algorithm 2** Entropy-based facet Assignment

---

**Require:** Feature matrix $\tilde{\mathbf{f}}$, labels $y$, constraint matrix $\phi$, thresholds $\tilde{\xi}$
**Require:** Valid facet threshold $\tau$, entropy threshold $h_{\text{target}}$, max attempts $T$
1: $\mathcal{U} \leftarrow \{i : y_i = -1\}$  ➤ Set of unsafe examples
2: $k_i \leftarrow \arg\max_k(\phi_k^\top \mathbf{f}_i - \tilde{\xi}_k)$ for all $i \in \mathcal{U}$  ➤ Initial assignments
3: $H \leftarrow \text{ComputeEntropy}(\{k_i\}_{i \in \mathcal{U}})$
4: **while** $H < h_{\text{target}}$ and attempts $< T$ **do**
5:    Sample $i$ randomly from $\mathcal{U}$
6:    $\mathcal{K}_{\text{valid}} \leftarrow \{k : \phi_k^\top \tilde{\mathbf{f}}_i - \tilde{\xi}_k > \tau\}$
7:    Sample $k_{\text{new}}$ randomly from $\mathcal{K}_{\text{valid}}$
8:    $H_{\text{new}} \leftarrow \text{ComputeEntropy}(\{k_i\}_{i \in \mathcal{U}} \cup \{k_{\text{new}}\})$
9:    **if** $H_{\text{new}} > H$ **then**
10:       $k_i \leftarrow k_{\text{new}}$
11:       $H \leftarrow H_{\text{new}}$
12:    **end if**
13: **end while**
**output** Facet assignments $\{k_i\}_{i \in \mathcal{U}}$

---

The algorithm iteratively attempts to increase assignment entropy by randomly selecting alternative facets from $\mathcal{K}_{\text{valid}}$ while maintaining strong constraint violations. The target entropy threshold $h_{\text{target}}$ is heuristically set to $\frac{1}{2} \log_2 K$, where $K$ is the total number of facets. $\tau$ is set to 0. This approach ensures diverse facet utilization across unsafe examples while preserving the effectiveness of the safety constraints.

# B. Steering via Lagrangian Relaxation

Building on the learned polytope facets, SaP implements directional steering in the model's representation space to constrain generation while minimizing disruption to the model's capabilities. This approach enables fine-grained safety control through local activation editing, avoiding the need for model fine-tuning or prompt engineering.

One way to solve the constrained problem in Equation (2) is through a Lagrangian relaxation. To this end, in practice, we perform 100 gradient steps on the following loss function

$$\min_{\boldsymbol{h}} \|\bar{\pi}_l(\boldsymbol{x}) - \boldsymbol{h}\|_1 + \lambda_{\text{safe}} \sum_{k=1}^{K} \left[\phi_k^\top \mathrm{E}_C(\boldsymbol{h}) - \tilde{\xi}_k\right]_- + \lambda_{\text{unsafe}} \sum_{k=1}^{K} \left[\phi_k^\top \mathrm{E}_C(\boldsymbol{h}) - \tilde{\xi}_k\right]_+$$

where $\boldsymbol{h}$ is the edited activation, $\mathrm{E}_C(\boldsymbol{h})$ is its corresponding encoded feature, $[\cdot]_+ = \max(0, \cdot)$ denotes the positive part, and $[\cdot]_- = \min(0, \cdot)$ denotes the negative part. The hyperparameters $\lambda_{\text{safe}}$ and $\lambda_{\text{unsafe}}$ control the penalties for negative and positive violations respectively, allowing for asymmetric treatment of constraint violations.

# C. Harmbench Training Setup and Results

## C.1. Harmbench Training Setup

In Harmbench (Mazeika et al., 2024), we save the per-token representation at layer 20 during the original model output generation (with attacked inputs). The total training dataset size is `num_sentences × tokens_per_sentence`. If an output sentence is marked as unsafe, we label all token representations in the sentence as unsafe, and vice versa. For each model (Llama, Mistral, Qwen), we use its default system prompt template when inferencing both the original and the steered model.

For the polytope training phase, we use the Adam optimizer with a learning rate of $10^{-2}$ and batch size of 128. The feature extractor projects hidden states to a 16,384-dimensional space followed by ReLU activation. The loss function uses an entropy weight 1.0 and feature L1 regularization weight $\lambda_\phi = 1.0$. The margin parameter $\kappa$ varies across model architectures: 60.0 for Llama-2 7B, 5.0 for Ministral 8B, and 30.0 for Qwen2 1.5B, reflecting different geometric requirements in their respective representation spaces.

During the steering phase, we apply hidden state optimization at layer 20 with model-specific configurations. For Llama-2 7B, we set $\lambda_{\text{unsafe}} = 4.0$ and $\lambda_{\text{safe}} = 10^{-4}$ for the optimization objective. Ministral 8B uses $\lambda_{\text{unsafe}} = 0.25$ without safety violation penalty ($\lambda_{\text{safe}} = 0$). For Qwen2 1.5B, we set $\lambda_{\text{unsafe}} = 10.0$ and $\lambda_{\text{safe}} = 5000$. All models are quantized to 16-bit precision (float16 for Llama-2 and bfloat16 for Ministral and Qwen) to reduce memory requirements while maintaining numerical stability.

## C.2. Harmbench Main Results

Figure 2 is created by aggregating the average of each defense method's performance over all 7 attack algorithms. We present detailed Llama-2 7B results in Table 2 and Table 3, Ministral-8B results in Table 4 and Table 5, and Qwen2-1.5B results in Table 6 and Table 7. We perform experiments in the same setup with prompt-based baselines Self Reminder (Xie et al., 2023), Response Check (Wang et al., 2024), and SmoothLLM (Robey et al., 2024), implemented in the BackTranslation code base (Wang et al., 2024).

We also implemented rejection sampling, which uses the polytope classifier. Instead of steering when a token is detected unsafe, it immediately modifies the current token output to be "Sorry". It empirically works well for defense, but might result in over-rejecting reasonable requests, as shown in the MMLU results (Table 8). All methods' results (mean ± standard deviation) are obtained over 5 seeds. Table 10 shows results over 7 attack algorithms for each model, when trained with and without the concept encoder.

Table 2: Defense comparison for Llama2-7B across different attack methods. Lower values indicate better defense effectiveness (0 means perfect defense). Each method is evaluated over 5 seeds.

| Attack Method | Original | Steering | Rejection | Response Check | Self Reminder | SmoothLLM |
|---|---|---|---|---|---|---|
| AutoDAN | 1.77 | 0.00 ± 0.00 | 0.00 ± 0.00 | 0.51 ± 0.00 | 0.00 ± 0.00 | 2.33 ± 0.33 |
| AutoPrompt | 16.5 | 0.00 ± 0.00 | 0.05 ± 0.11 | 8.25 ± 0.00 | 1.25 ± 0.00 | 10.30 ± 1.58 |
| DirectRequest | 1.0 | 0.00 ± 0.00 | 0.05 ± 0.11 | 0.50 ± 0.00 | 0.00 ± 0.00 | 6.50 ± 0.64 |
| GBDA | 0 | 0.00 ± 0.00 | 0.00 ± 0.00 | 0.06 ± 0.04 | 0.00 ± 0.00 | 4.21 ± 0.39 |
| GCG | 30 | 0.40 ± 0.29 | 0.10 ± 0.22 | 13.00 ± 0.00 | 2.50 ± 0.00 | 12.10 ± 0.68 |
| HumanJailbreaks | 0.5 | 0.00 ± 0.00 | 0.01 ± 0.02 | 0.09 ± 0.01 | 0.05 ± 0.00 | 2.19 ± 0.45 |
| PEZ | 0 | 0.00 ± 0.00 | 0.00 ± 0.00 | 0.00 ± 0.00 | 0.00 ± 0.00 | 4.03 ± 0.48 |
| UAT | 6.5 | 0.00 ± 0.00 | 0.00 ± 0.00 | 2.75 ± 0.00 | 0.50 ± 0.00 | 8.45 ± 0.65 |
| AdaptiveAttack | 60 | 2.00 ± 0.00 | 0.00 ± 0.00 | 0.00 ± 0.00 | 2.00 ± 0.00 | 2.04 ± 0.00 |

Table 3: MLP defense comparison for Llama2-7B across different attack methods, evaluated over 5 seeds. Lower values indicate better defense effectiveness.

| Attack Method | Original | MLP | MLP+5 |
|---|---|---|---|
| AutoDAN | 1.77 | 0.25 ± 0.00 | 0.25 ± 0.00 |
| AutoPrompt | 16.5 | 12.80 ± 0.27 | 12.85 ± 0.14 |
| DirectRequest | 1.0 | 0.30 ± 0.11 | 0.25 ± 0.00 |
| GBDA | 0 | 0.00 ± 0.00 | 0.00 ± 0.00 |
| GCG | 30 | 27.10 ± 0.29 | 26.90 ± 0.29 |
| HumanJailbreaks | 0.5 | 0.34 ± 0.26 | 0.24 ± 0.02 |
| PEZ | 0 | 0.00 ± 0.00 | 0.00 ± 0.00 |
| UAT | 6.5 | 4.75 ± 0.18 | 4.75 ± 0.00 |
| AdaptiveAttack | 60 | 0.00 ± 0.00 | 0.00 ± 0.00 |

Table 4: Defense comparison for Mistral-8B across different attack methods. Lower values indicate better defense effectiveness (0 means perfect defense). Each method is evaluated over 5 seeds.

| Attack Method | Original | Steering | Rejection | Response Check | Self Reminder | SmoothLLM |
|---|---|---|---|---|---|---|
| AutoDAN | 66.75 | 1.35 ± 0.49 | 0.00 ± 0.00 | 1.50 ± 0.00 | 11.50 ± 0.00 | 19.38 ± 1.94 |
| AutoPrompt | 48.3 | 2.79 ± 1.64 | 0.51 ± 0.25 | 7.88 ± 0.00 | 13.94 ± 0.00 | 40.36 ± 2.63 |
| DirectRequest | 42.8 | 1.20 ± 0.54 | 0.33 ± 0.13 | 5.25 ± 0.00 | 2.50 ± 0.00 | 16.30 ± 2.18 |
| GBDA | 62.8 | 1.26 ± 1.10 | 0.10 ± 0.23 | 14.62 ± 4.37 | 6.75 ± 0.00 | 36.75 ± 8.71 |
| GCG | 67.5 | 1.90 ± 1.10 | 0.08 ± 0.13 | 5.75 ± 0.00 | 26.50 ± 0.00 | 32.35 ± 2.00 |
| HumanJailbreaks | 39.4 | 1.15 ± 0.68 | 0.01 ± 0.02 | 2.36 ± 0.13 | 9.65 ± 0.01 | 17.41 ± 0.88 |
| PEZ | 33.6 | 1.14 ± 0.40 | 0.00 ± 0.00 | 11.38 ± 4.79 | 1.97 ± 0.00 | 27.00 ± 10.63 |
| UAT | 40.8 | 2.45 ± 1.04 | 0.54 ± 0.25 | 6.25 ± 0.00 | 4.50 ± 0.00 | 24.65 ± 3.09 |
| AdaptiveAttack | 100 | 14.00 ± 0.00 | 0.00 ± 0.00 | 97.33 ± 0.94 | 90.00 ± 0.00 | 97.33 ± 0.94 |

Table 5: MLP defense comparison for Ministral-8B across different attack methods, evaluated over 5 seeds. Lower values indicate better defense effectiveness.

| Attack Method | Original | MLP | MLP+5 |
|---|---|---|---|
| AutoDAN | 66.75 | 16.90 ± 5.14 | 17.80 ± 1.34 |
| AutoPrompt | 48.3 | 74.30 ± 2.04 | 73.58 ± 1.58 |
| DirectRequest | 42.8 | 20.50 ± 1.06 | 20.35 ± 1.29 |
| GBDA | 62.8 | 66.31 ± 22.82 | 80.91 ± 3.12 |
| GCG | 67.5 | 60.35 ± 0.63 | 61.10 ± 0.82 |
| HumanJailbreaks | 39.4 | 27.96 ± 8.03 | 25.62 ± 1.75 |
| PEZ | 33.6 | 32.72 ± 12.77 | 32.81 ± 15.03 |
| UAT | 40.8 | 49.90 ± 1.07 | 49.25 ± 1.38 |
| AdaptiveAttack | 100 | 16.00 ± 0.00 | 16.00 ± 0.00 |

# D. BeaverTails Training Setup and Results

## D.1. Mutual Information Calculation Details

For BeaverTails (Ji et al., 2023), the Mutual Information (MI) between each safety category and each facet's violations are computed using scikit-learn's mutual_information_score function. We compare the facets training from two polytopes, both share the same set of hyperparameters, and the only difference is whether the Concept Encoder is included in the training. They are both trained with 50 facets, trained on the full BeaverTails dataset over 20 epochs, optimized with Adam optimizer with learning rate 0.01, batch size 128, and $\lambda_\phi$ 0.01. The polytope trained with Concept Encoder has an encoded feature dimension 16384, $\lambda_{\tilde{\mathbf{f}}}$ 1.0, and a margin 2.0.

Table 6: Defense comparison for Qwen-1.5B across different attack methods. Lower values indicate better defense effectiveness (0 means perfect defense). Each method is evaluated over 5 seeds.

| Attack Method | Original | Steering | Rejection | Response Check | Self Reminder | SmoothLLM |
|---|---|---|---|---|---|---|
| AutoDAN | 45.00 | 0.80 ± 0.82 | 0.67 ± 0.00 | 38.25 ± 0.00 | 30.50 ± 0.00 | 6.85 ± 1.27 |
| AutoPrompt | 17.90 | 1.16 ± 0.55 | 2.83 ± 1.09 | 23.74 ± 0.00 | 3.54 ± 0.00 | 6.77 ± 1.20 |
| DirectRequest | 13.25 | 0.15 ± 0.22 | 0.60 ± 0.29 | 5.00 ± 0.00 | 2.00 ± 0.00 | 5.60 ± 1.04 |
| GBDA | 19.45 | 1.42 ± 0.87 | 3.02 ± 0.52 | 20.63 ± 2.41 | 3.40 ± 0.00 | 8.54 ± 0.63 |
| GCG | 25.00 | 1.00 ± 0.69 | 2.64 ± 2.43 | 45.00 ± 0.00 | 22.86 ± 0.00 | 11.43 ± 1.16 |
| HumanJailbreaks | 4.14 | 0.18 ± 0.13 | 0.28 ± 0.23 | 5.40 ± 0.00 | 4.00 ± 0.00 | 4.42 ± 0.50 |
| PEZ | 10.90 | 0.45 ± 0.44 | 1.06 ± 0.74 | 4.85 ± 0.00 | 0.75 ± 0.00 | 4.63 ± 0.48 |
| UAT | 12.50 | 1.15 ± 0.52 | 2.70 ± 1.14 | 14.75 ± 0.00 | 1.75 ± 0.00 | 7.30 ± 1.56 |
| AdaptiveAttack | 100 | 100.00 ± 0.00 | 90.00 ± 0.00 | 100.00 ± 0.00 | 100.00 ± 0.00 | 100.00 ± 0.00 |

Table 7: MLP defense comparison for Qwen-1.5B across different attack methods, evaluated over 5 seeds. Lower values indicate better defense effectiveness.

| Attack Method | Original | MLP | MLP+5 |
|---|---|---|---|
| AutoDAN | 45.00 | 3.75 ± 1.56 | 12.85 ± 14.36 |
| AutoPrompt | 17.90 | 8.23 ± 1.62 | 14.29 ± 12.29 |
| DirectRequest | 13.25 | 3.10 ± 0.63 | 4.00 ± 2.67 |
| GBDA | 19.45 | 9.92 ± 1.44 | 16.80 ± 11.80 |
| GCG | 25.00 | 10.57 ± 3.35 | 18.43 ± 21.13 |
| HumanJailbreaks | 4.14 | 2.26 ± 0.55 | 4.17 ± 4.38 |
| PEZ | 10.9 | 2.81 ± 0.50 | 5.07 ± 2.21 |
| UAT | 12.5 | 6.75 ± 1.08 | 10.95 ± 6.36 |
| AdaptiveAttack | 100 | 100.00 ± 0.00 | 100.00 ± 0.00 |

Table 8: MMLU accuracy (%) comparison across different defense methods. Results show that most defense methods maintain original model performance, with some exceptions for rejection and self-reminder strategies.

| Method | Llama2-7B | Ministral-8B | Qwen2-1.5B |
|---|---|---|---|
| Original | 45.8 | 63.4 | 52.7 |
| Steering | 45.7 | 63.3 | 52.7 |
| Rejection | 44.6 | 28.5 | 51.2 |
| ICL | 45.8 | 65.1 | 55.4 |
| Response Check | 45.8 | 65.1 | 55.4 |
| Self Reminder | 45.8 | 22.9 | 22.9 |
| SmoothLLM | 45.8 | 65.1 | 55.4 |
| MLP | 45.8 | 63.4 | 55.4 |
| MLP+5 | 45.8 | 63.4 | 55.4 |

## D.2. BeaverTails Training Hyperparameters

We conduct polytope training on all BeaverTails classes. We first obtain the (human question, model answer) pairs from the dataset, formatting the input as `f"{human_question}\n{model_answer}"` using Python f-string without any other prompts. We observe that this works better for classifying safety concepts than other basic types of input prompting. We then obtain the activation for each sentence at a model's layer $l$ at the sentence's last word.

The polytope model is optimized using Adam with a learning rate of $10^{-2}$ and batch size of 128. For facet assignment, we use a valid facet threshold $\tau = 0.0$. The concept encoder uses a feature dimension of 16,384. The training runs for 20 epochs.

Table 9: Model performance across turns in MT-Bench. There are fluctuations in LLM-based judges' scores, but all models produce coherent and high-quality sentences, without signs of rejections.

| Model | First Turn | Second Turn | Average |
|---|---|---|---|
| Llama2-7B | 6.54 | 5.46 | 6.00 |
| Llama2-7B + SaP | 6.98 | 6.21 | 6.59 |
| Ministral-8B | 7.55 | 6.98 | 7.26 |
| Ministral-8B + SaP | 9.01 | 7.64 | 8.33 |
| Qwen-1.5B | 6.90 | 5.15 | 6.03 |
| Qwen-1.5B + SaP | 6.29 | 4.70 | 5.49 |

Table 10: Attack success rate comparison between models, with and without Concept Encoder. Results show mean ± standard deviation across different attack methods.

| | Llama-2 7B | | Ministral-8B | | Qwen2 1.5B | |
|---|---|---|---|---|---|---|
| Method | Without CE | With CE | Without CE | With CE | Without CE | With CE |
| AutoPrompt | 10.7 ± 0.0 | 0.00 ± 0.00 | 73.3 ± 0.0 | 2.79 ± 1.64 | 0.45 ± 0.33 | 1.16 ± 0.55 |
| DirectRequest | 0.0 ± 0.0 | 0.00 ± 0.00 | 21.5 ± 0.0 | 1.20 ± 0.54 | 0.15 ± 0.14 | 0.15 ± 0.22 |
| GBDA | 0.0 ± 0.0 | 0.00 ± 0.00 | 82.3 ± 0.0 | 1.26 ± 1.10 | 0.71 ± 0.28 | 1.42 ± 0.87 |
| GCG | 26.5 ± 0.0 | 0.40 ± 0.29 | 60.8 ± 0.0 | 1.90 ± 1.10 | 0.57 ± 0.54 | 1.00 ± 0.69 |
| HumanJailbreaks | 0.3 ± 0.0 | 0.00 ± 0.00 | 24.7 ± 0.0 | 1.15 ± 0.68 | 0.08 ± 0.08 | 0.18 ± 0.13 |
| PEZ | 0.0 ± 0.0 | 0.00 ± 0.00 | 48.0 ± 0.0 | 1.14 ± 0.40 | 0.32 ± 0.07 | 0.45 ± 0.44 |
| UAT | 4.0 ± 0.0 | 0.00 ± 0.00 | 50.2 ± 0.0 | 2.45 ± 1.04 | 0.45 ± 0.33 | 1.15 ± 0.52 |

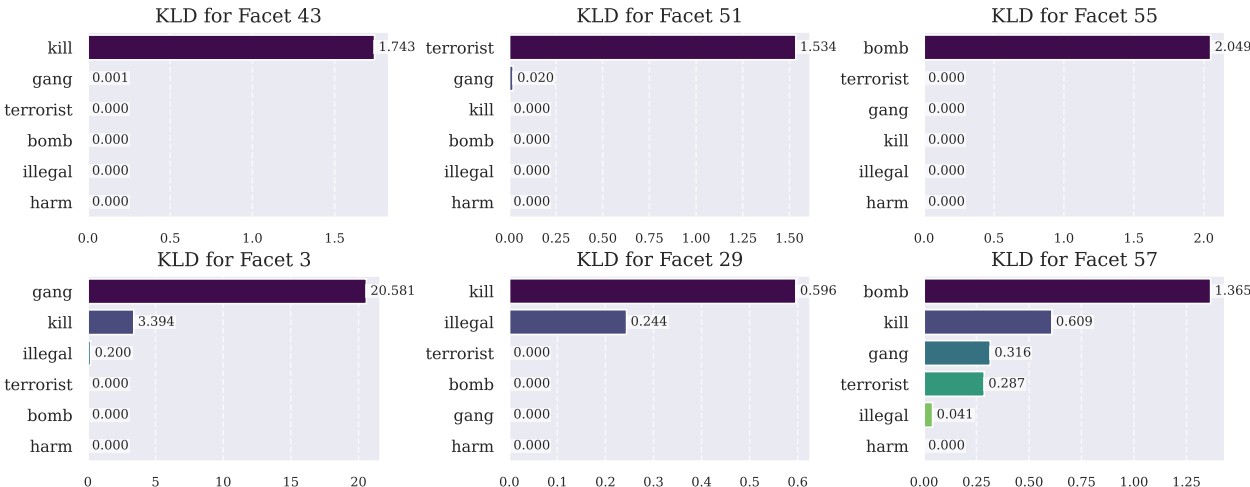

Figure 7: KL divergence analysis revealing semantic specialization of SaP's facets on BeaverTail's terrorism dataset. Facets can activate at one single concept (top row plots), or multiple concepts (bottom row plots)

We empirically find that margin parameter $\kappa$ and the polytope sparsity weight $\lambda_\phi$ significantly influence model performance. Different safety categories and model architectures require different hyperparameter values for optimal performance. Table 11 shows hyperparameters for Llama2-7B, and Table 12 and 13 show hyperparameters for Ministral-8B and Qwen2-1.5B respectively.

Table 11: Category-specific hyperparameters for BeaverTails for Llama2-7B. The last row "All data" shows the hyperparameters used to train all BeaverTails data.

| Category | Margin | $\lambda_\phi$ |
|---|---|---|
| Animal abuse | 0.5 | 0.01 |
| Child abuse | 0.7 | 0.01 |
| Politics | 3.0 | 0.0001 |
| Discrimination | 2.0 | 0.0001 |
| Drug abuse | 1.0 | 0.01 |
| Financial crime | 2.0 | 0.01 |
| Hate speech | 10.0 | 0.0001 |
| Misinformation | 0.8 | 0.0001 |
| Unethical behavior | 20.0 | 0.0001 |
| Privacy violation | 1.0 | 0.01 |
| Self-harm | 1.0 | 0.0001 |
| Adult content | 1.0 | 0.01 |
| Terrorism | 1.0 | 0.01 |
| Violence | 20.0 | 0.0001 |
| All data | 20.0 | 0.01 |

Table 12: Category-specific hyperparameters for BeaverTails for Ministral-8B. The last row "All data" shows the hyperparameters used to train all BeaverTails data.

| Category | Margin | $\lambda_\phi$ |
|---|---|---|
| Animal abuse | 1.0 | 0.0001 |
| Child abuse | 0.8 | 0.0001 |
| Politics | 10.0 | 0.0001 |
| Discrimination | 10.0 | 0.0001 |
| Drug abuse | 2.0 | 0.0001 |
| Financial crime | 2.0 | 0.0001 |
| Hate speech | 1.0 | 0.0001 |
| Misinformation | 10.0 | 0.0001 |
| Unethical behavior | 15.0 | 0.0001 |
| Privacy violation | 2.0 | 0.0001 |
| Self-harm | 2.0 | 0.01 |
| Adult content | 0.6 | 0.01 |
| Terrorism | 0.6 | 0.01 |
| Violence | 2.0 | 0.01 |
| All data | 20.0 | 0.0001 |

### D.3. BeaverTails Results

We present test accuracy results across different configurations of facets of Llama2-7B, Ministral-8B, and Qwen2-1.5B in Tables 14, 15, and 16 respectively. For each category, we additionally run learning on a multi-layer perceptron (MLP) classifier using the same representation inputs. For fair comparison, the MLP is defined to have a similar model size with the polytope trained for each safety category. The MLP architecture has two layers and a classification head:

- The first layer is $\text{ReLU}(W_1 \bar{\pi}_l(\boldsymbol{x}) + b_1)$, with $\bar{\pi}_l(\boldsymbol{x})$ being the LLM activation at layer $l$ (i.e., the same representation input as the one used in polytope training). This layer has the same number of parameters as the concept encoder, with $W_1 \in \mathbb{R}^{16384 \times d_l}$, $b_1 \in \mathbb{R}^{16384}$.

- We then add a second layer $\text{ReLU}(W_2 x_1 + b_2)$, with $W_2 \in \mathbb{R}^{n_{\text{facet}} \times 16384}$, with $x_1$ being the output of the first layer, $b_1 \in \mathbb{R}^{n_{\text{facet}}}$ matching the parameter size of the polytope $\phi$ and $\tilde{\xi}$.

Table 13: Category-specific hyperparameters for BeaverTails for Qwen-1.5B. The last row "All data" shows the hyperparameters used to train all BeaverTails data.

| Category | Margin | $\lambda_\phi$ |
|---|---|---|
| Animal abuse | 15.0 | 0.0001 |
| Child abuse | 2.0 | 0.01 |
| Politics | 20.0 | 0.0001 |
| Discrimination | 10.0 | 0.0001 |
| Drug abuse | 15.0 | 0.0001 |
| Financial crime | 10.0 | 0.0001 |
| Hate speech | 0.8 | 0.01 |
| Misinformation | 0.8 | 0.01 |
| Unethical behavior | 15.0 | 0.0001 |
| Privacy violation | 2.0 | 0.0001 |
| Self-harm | 2.0 | 0.0001 |
| Adult content | 3.0 | 0.0001 |
| Terrorism | 0.9 | 0.0001 |
| Violence | 10.0 | 0.0001 |
| All data | 10.0 | 0.01 |

- To perform classification, MLP has an additional classification head defined by $\text{Softmax}(W_3 x_2 + b_3)$, where $x_2$ is the second layer output, $W_3 \in \mathbb{R}^{2 \times n_{\text{facet}}}$, $b_3 \in \mathbb{R}^2$. It outputs the model's estimated probability whether an input is safe or unsafe (i.e., 2 classes). Hence, for each category, the MLP has $n_{\text{facet}} \times 2 + 2$ more parameters than the SaP polytope.

The MLP is trained using cross-entropy loss and an Adam optimizer with learning rate 0.0001. We present the results in Table 17. More KL divergence analysis on SaP's facets are presented in Table 7.

Table 14: Test accuracy (%) comparison across different numbers of facets for each category in Llama2-7B. Results show mean ± standard deviation. Bold numbers indicate the selected configuration in our final model.

| Category | 1 facet | 10 facets | 20 facets | 30 facets | 40 facets | 50 facets | 60 facets |
|---|---|---|---|---|---|---|---|
| Animal abuse | 60.0 ± 18.0 | 92.0 ± 3.0 | 93.0 ± 1.4 | 92.0 ± 3.0 | 91.0 ± 4.0 | **93.0 ± 1.0** | 89.0 ± 2.0 |
| Child abuse | 85.0 ± 8.0 | **91.8 ± 2.2** | 89.0 ± 4.0 | 88.0 ± 1.0 | 88.0 ± 2.0 | 87.0 ± 2.0 | 89.0 ± 3.0 |
| Politics | 82.0 ± 3.0 | 82.0 ± 1.0 | 82.0 ± 2.0 | **84.1 ± 0.5** | 80.0 ± 2.0 | 81.0 ± 1.0 | 80.0 ± 2.0 |
| Discrimination | 86.0 ± 2.0 | 81.0 ± 5.0 | 85.0 ± 2.0 | 85.0 ± 2.0 | **92.8 ± 1.8** | 82.0 ± 4.0 | 82.0 ± 3.0 |
| Drug abuse | **93.7 ± 0.8** | 89.0 ± 7.0 | 92.0 ± 1.0 | 90.0 ± 2.0 | 90.0 ± 3.0 | 91.0 ± 1.0 | 92.0 ± 1.0 |
| Financial crime | **92.6 ± 1.0** | 92.0 ± 1.0 | 90.0 ± 2.0 | 87.0 ± 6.0 | 88.0 ± 1.0 | 91.0 ± 1.0 | 90.0 ± 2.0 |
| Hate speech | 80.0 ± 8.0 | **88.3 ± 0.7** | 82.0 ± 3.0 | 80.0 ± 6.0 | 84.0 ± 2.0 | 80.0 ± 9.0 | 83.0 ± 4.0 |
| Misinformation | 55.0 ± 9.0 | **75.5 ± 1.1** | 71.0 ± 7.0 | 72.0 ± 2.0 | 67.0 ± 6.0 | 69.0 ± 2.0 | 70.0 ± 2.0 |
| Unethical behavior | 83.0 ± 6.0 | 85.0 ± 1.0 | 85.0 ± 1.0 | 85.0 ± 1.0 | **87.5 ± 0.6** | 84.0 ± 1.0 | 81.0 ± 4.0 |
| Privacy violation | 89.0 ± 5.0 | **94.0 ± 2.0** | 92.0 ± 1.0 | 92.0 ± 2.0 | 93.0 ± 1.0 | 91.0 ± 2.0 | 92.0 ± 1.0 |
| Self-harm | 90.0 ± 3.0 | **93.0 ± 2.0** | 91.0 ± 2.0 | 93.0 ± 3.0 | 92.0 ± 3.0 | 92.0 ± 2.0 | 92.5 ± 1.6 |
| Adult content | 90.0 ± 3.0 | **93.5 ± 1.0** | 91.0 ± 2.0 | 90.0 ± 2.0 | 89.0 ± 2.0 | 89.0 ± 2.0 | 86.0 ± 3.0 |
| Terrorism | 90.0 ± 3.0 | 91.0 ± 2.0 | 89.1 ± 1.1 | **92.0 ± 1.0** | 88.0 ± 5.0 | 90.0 ± 3.0 | 85.0 ± 9.0 |
| Violence | 87.0 ± 2.0 | 87.0 ± 2.0 | 87.0 ± 3.0 | 86.0 ± 3.0 | **90.9 ± 0.5** | 86.0 ± 3.0 | 86.0 ± 3.0 |

Table 15: Test accuracy (%) comparison across different numbers of facets for each category in Ministral-8B. Results show mean ± standard deviation. Bold numbers indicate the selected configuration in our final model.

| Category | 1 facet | 10 facets | 20 facets | 30 facets | 40 facets | 50 facets | 60 facets |
|---|---|---|---|---|---|---|---|
| Animal abuse | 50.0 ± 0.0 | 80.0 ± 20.0 | **93.6 ± 1.4** | 88.0 ± 6.0 | 80.0 ± 17.0 | 61.0 ± 11.0 | 84.0 ± 4.0 |
| Child abuse | 86.0 ± 6.0 | **89.9 ± 1.3** | 85.0 ± 2.0 | 86.0 ± 4.0 | 88.0 ± 1.0 | 88.0 ± 2.0 | 85.0 ± 2.0 |
| Politics | **86.9 ± 0.4** | 76.0 ± 5.0 | 62.0 ± 13.0 | 57.0 ± 8.0 | 76.0 ± 6.0 | 73.0 ± 8.0 | 65.0 ± 17.0 |
| Discrimination | 82.0 ± 2.0 | 76.0 ± 11.0 | 61.0 ± 13.0 | 78.0 ± 6.0 | 70.0 ± 9.0 | **87.2 ± 0.3** | 69.0 ± 13.0 |
| Drug abuse | **93.7 ± 0.8** | 68.0 ± 14.0 | 84.0 ± 14.0 | 73.0 ± 16.0 | 73.0 ± 20.0 | 82.0 ± 19.0 | 88.0 ± 7.0 |
| Financial crime | **92.6 ± 1.0** | 91.0 ± 1.0 | 91.0 ± 1.0 | 91.0 ± 1.0 | 90.0 ± 2.0 | 90.0 ± 1.0 | 91.0 ± 1.0 |
| Hate speech | 86.0 ± 4.0 | **88.3 ± 0.7** | 82.0 ± 7.0 | 85.0 ± 3.0 | 87.0 ± 2.0 | 81.0 ± 4.0 | 81.0 ± 3.0 |
| Misinformation | 56.0 ± 12.0 | **75.5 ± 1.1** | 73.0 ± 2.0 | 70.0 ± 7.0 | 71.0 ± 2.0 | 72.0 ± 3.0 | 68.0 ± 8.0 |
| Unethical behavior | 85.0 ± 2.0 | 85.0 ± 1.0 | **87.8 ± 0.6** | 86.0 ± 0.0 | 84.0 ± 1.0 | 82.0 ± 4.0 | 82.0 ± 5.0 |
| Privacy violation | 84.0 ± 10.0 | **94.0 ± 2.0** | 93.0 ± 2.0 | 91.0 ± 3.0 | 91.0 ± 5.0 | 93.0 ± 2.0 | 91.0 ± 4.0 |
| Self-harm | 91.0 ± 2.0 | 91.0 ± 6.0 | 92.5 ± 1.6 | 91.0 ± 2.0 | 92.0 ± 2.0 | 89.0 ± 3.0 | **93.0 ± 2.0** |
| Adult content | 92.0 ± 3.0 | **93.5 ± 1.0** | 91.0 ± 3.0 | 93.0 ± 2.0 | 93.0 ± 0.0 | 92.0 ± 2.0 | 91.0 ± 2.0 |
| Terrorism | 87.0 ± 5.0 | 89.0 ± 1.0 | **89.1 ± 1.1** | 88.0 ± 1.0 | 89.0 ± 1.0 | 88.0 ± 1.0 | 87.0 ± 2.0 |
| Violence | 88.0 ± 4.0 | 90.0 ± 1.0 | 88.0 ± 3.0 | **90.9 ± 0.5** | 87.0 ± 2.0 | 86.0 ± 4.0 | 89.0 ± 1.0 |

Table 16: Test accuracy (%) comparison across different numbers of facets for each category in Qwen2-1.5B. Results show mean ± standard deviation. Bold numbers indicate the selected configuration in our final model.

| Category | 1 facet | 10 facets | 20 facets | 30 facets | 40 facets | 50 facets | 60 facets |
|---|---|---|---|---|---|---|---|
| Animal abuse | **93.5 ± 1.1** | 89.0 ± 6.0 | 88.0 ± 5.0 | 84.0 ± 7.0 | 84.0 ± 7.0 | 86.0 ± 4.0 | 83.0 ± 10.0 |
| Child abuse | 83.0 ± 8.0 | **86.2 ± 2.2** | 85.0 ± 2.0 | 83.0 ± 3.0 | 84.0 ± 3.0 | 85.0 ± 2.0 | 82.0 ± 4.0 |
| Politics | 81.0 ± 3.0 | 80.0 ± 6.0 | 77.0 ± 8.0 | 77.0 ± 6.0 | 78.0 ± 2.0 | **83.3 ± 1.0** | 77.0 ± 7.0 |
| Discrimination | 76.0 ± 6.0 | 76.0 ± 5.0 | 78.0 ± 3.0 | **85.3 ± 0.4** | 76.0 ± 10.0 | 80.0 ± 4.0 | 81.0 ± 1.0 |
| Drug abuse | 90.0 ± 3.0 | 88.0 ± 6.0 | **92.6 ± 0.9** | 87.0 ± 8.0 | 90.0 ± 1.0 | 89.0 ± 4.0 | 89.0 ± 5.0 |
| Financial crime | 90.0 ± 3.0 | 88.0 ± 3.0 | **91.0 ± 1.0** | 89.0 ± 2.0 | 89.0 ± 4.0 | 89.0 ± 1.0 | 84.8 ± 1.2 |
| Hate speech | 82.0 ± 5.0 | 84.8 ± 1.2 | 82.0 ± 3.0 | 83.0 ± 5.0 | 85.0 ± 2.0 | 84.0 ± 1.0 | **85.0 ± 1.0** |
| Misinformation | 50.0 ± 0.0 | 63.0 ± 7.0 | 71.0 ± 2.0 | **71.5 ± 2.4** | 69.0 ± 1.0 | 67.0 ± 5.0 | 68.0 ± 5.0 |
| Unethical behavior | 81.0 ± 7.0 | 83.0 ± 3.0 | **86.3 ± 0.7** | 82.0 ± 4.0 | 80.0 ± 3.0 | 83.0 ± 1.0 | 82.0 ± 1.0 |
| Privacy violation | 90.0 ± 3.0 | 88.0 ± 4.0 | 90.0 ± 2.0 | 89.0 ± 2.0 | **93.1 ± 0.4** | 91.0 ± 1.0 | 88.0 ± 2.0 |
| Self-harm | 77.0 ± 14.0 | 86.0 ± 4.0 | **90.4 ± 1.6** | 88.0 ± 3.0 | 88.0 ± 2.0 | 88.0 ± 2.0 | 87.0 ± 2.0 |
| Adult content | 89.0 ± 1.0 | **91.5 ± 0.5** | 85.0 ± 3.0 | 88.0 ± 4.0 | 86.0 ± 2.0 | 88.0 ± 2.0 | 86.0 ± 5.0 |
| Terrorism | 85.0 ± 6.0 | 90.0 ± 1.0 | 88.0 ± 3.0 | 88.0 ± 2.0 | 88.0 ± 2.0 | **89.7 ± 1.6** | 89.0 ± 2.0 |
| Violence | 85.0 ± 3.0 | 81.0 ± 9.0 | 72.0 ± 15.0 | **88.2 ± 1.3** | 84.0 ± 2.0 | 86.0 ± 2.0 | 84.0 ± 3.0 |

Table 17: Classification accuracy (%) on the BeaverTails dataset. Results show mean ± standard deviation across different safety categories. MLP uses an architecture with a similar amount of Safety Polytope parameters, but has additional parameters in the classification layer.

| Category | Llama2-7B | | | Ministral-8B | | | Qwen2-1.5B | | |
|---|---|---|---|---|---|---|---|---|---|
| | 1 facet | SaP | MLP | 1 facet | SaP | MLP | 1 facet | SaP | MLP |
| Animal abuse | 60.0 ± 18.0 | 93.0 ± 1.0 | 96.0 | 50.0 ± 0.0 | 93.6 ± 1.4 | 95.5 | 93.5 ± 1.1 | 93.5 ± 1.1 | 94.8 |
| Child abuse | 85.0 ± 8.0 | 91.8 ± 2.2 | 92.8 | 86.0 ± 6.0 | 89.9 ± 1.3 | 91.7 | 83.0 ± 8.0 | 86.2 ± 2.2 | 87.6 |
| Politics | 82.0 ± 3.0 | 84.1 ± 0.5 | 84.1 | 86.9 ± 0.4 | 86.9 ± 0.4 | 86.8 | 81.0 ± 3.0 | 83.3 ± 1.0 | 85.1 |
| Discrimination | 86.0 ± 2.0 | 92.8 ± 1.8 | 88.9 | 82.0 ± 2.0 | 87.2 ± 0.3 | 88.4 | 76.0 ± 6.0 | 85.3 ± 0.4 | 87.0 |
| Drug abuse | 93.7 ± 0.8 | 93.7 ± 0.8 | 95.1 | 93.7 ± 0.8 | 93.7 ± 0.8 | 95.9 | 90.0 ± 3.0 | 92.6 ± 0.9 | 93.6 |
| Financial crime | 92.6 ± 1.0 | 92.6 ± 1.0 | 94.0 | 92.6 ± 1.0 | 92.6 ± 1.0 | 94.7 | 90.0 ± 3.0 | 91.0 ± 1.0 | 93.5 |
| Hate speech | 80.0 ± 8.0 | 88.3 ± 0.7 | 88.5 | 86.0 ± 4.0 | 88.3 ± 0.7 | 88.3 | 82.0 ± 5.0 | 85.0 ± 2.0 | 86.9 |
| Misinformation | 55.0 ± 9.0 | 75.5 ± 1.1 | 71.7 | 56.0 ± 12.0 | 75.5 ± 1.1 | 77.6 | 50.0 ± 0.0 | 71.5 ± 2.4 | 74.5 |
| Unethical behavior | 83.0 ± 6.0 | 87.5 ± 0.6 | 88.8 | 85.0 ± 2.0 | 87.8 ± 0.6 | 88.3 | 81.0 ± 7.0 | 86.3 ± 0.7 | 86.8 |
| Privacy violation | 89.0 ± 5.0 | 94.0 ± 2.0 | 94.9 | 84.0 ± 10.0 | 94.0 ± 2.0 | 96.2 | 90.0 ± 3.0 | 93.1 ± 0.4 | 94.7 |
| Self-harm | 90.0 ± 3.0 | 93.0 ± 2.0 | 92.6 | 91.0 ± 2.0 | 93.0 ± 2.0 | 94.3 | 77.0 ± 14.0 | 90.4 ± 1.6 | 89.7 |
| Adult content | 90.0 ± 3.0 | 93.5 ± 1.0 | 92.7 | 92.0 ± 3.0 | 93.5 ± 1.0 | 94.1 | 89.0 ± 1.0 | 91.5 ± 0.5 | 91.3 |
| Terrorism | 90.0 ± 3.0 | 92.0 ± 1.0 | 92.7 | 87.0 ± 5.0 | 90.0 ± 1.0 | 90.0 | 85.0 ± 6.0 | 90.0 ± 1.0 | 92.3 |
| Violence | 87.0 ± 2.0 | 90.9 ± 0.5 | 91.6 | 88.0 ± 4.0 | 90.9 ± 0.5 | 91.9 | 85.0 ± 3.0 | 88.2 ± 1.3 | 90.6 |
| All data | 81.2 ± 0.2 | 82.2 ± 0.4 | 83.9 | 80.2 ± 0.5 | 82.2 ± 0.8 | 83.9 | 79.5 ± 0.8 | 80.3 ± 1.0 | 83.1 |

# E. Polytope Sample Complexity

We present the current theoretical results of learning a polytope. Note that these results make 3 important assumptions: **(i)** reliance on certain restricted encoder nature; **(ii)** perfect data separability by the polytope; **(iii)** the feasibility of finding a solution.

The following theorem is derived from (Gottlieb et al., 2021), which establishes the fat-shattering dimension of $\gamma$-fat $t$-polyhedra.

**Theorem E.1** (Theorem 2.2, Gottlieb et al. (2021)). *The class of $\gamma$-fat $t$-polyhedra in $\mathbb{R}^d$, with at most $t$ hyperplanes, has a fat-shattering dimension of order:*

$$D = O\left(\frac{t \log t}{\gamma^2}\right),$$

*where $t$ is the number of hyperplanes defining the polyhedron, and $\gamma$ is the margin.*

This result provides the fat-shattering dimension $D$ for our hypothesis class (the class of $\gamma$-fat polytopes), which we will use to derive the sample complexity.

The second theorem we rely on is a general result from Anthony & Bartlett (1999), which gives a sample complexity bound based on the fat-shattering dimension.

**Theorem E.2** (Theorem 19.1, Anthony & Bartlett (1999)). *Suppose that $F$ is a class of functions mapping from a domain $X$ into the real interval $[0, 1]$, and suppose also that $F$ has finite fat-shattering dimension. Let $\mathcal{A}$ be any approximate-SEM algorithm for $F$ and define, for $z \in Z^m$, $L(z) = \mathcal{A}(z, \epsilon_0/6)$, where $\epsilon_0 = 16/\sqrt{m}$. Then $L$ is a learning algorithm for $F$, and its sample complexity satisfies*

$$m(\epsilon, \delta) \leq \frac{256}{\epsilon^2}\left(18 \operatorname{fat}_F(\frac{\epsilon}{256}) \ln^2\left(\frac{128}{\epsilon}\right) + \ln\left(\frac{16}{\delta}\right)\right)$$

*for all $\epsilon, \delta > 0$.*

Using big-O notation, we simplify this to the following form:

$$m(\epsilon, \delta) = O\left(\frac{1}{\epsilon^2}\left(\operatorname{fat}_F(\epsilon) \log^2\left(\frac{1}{\epsilon}\right) + \log\left(\frac{1}{\delta}\right)\right)\right).$$

We now state and prove the main result, which gives the sample complexity bound for learning $\gamma$-fat polytopes.

**Corollary E.3** (Sample complexity of $\gamma$-fat polytopes). *Let $\mathcal{H}$ be the class of $\gamma$-fat $t$-polyhedra in $\mathbb{R}^d$. The sample complexity $m(\epsilon, \delta)$ required to learn a polytope from $\mathcal{H}$ with accuracy $\epsilon$ and confidence $1 - \delta$ is bounded by:*

$$m(\epsilon, \delta) = O\left(\frac{1}{\epsilon^2}\left(\frac{t \log t}{\gamma^2} \log^2\left(\frac{1}{\epsilon}\right) + \log\left(\frac{1}{\delta}\right)\right)\right).$$

*Proof.* By Theorem E.1, the fat-shattering dimension of the class of $\gamma$-fat $t$-polyhedra is:

$$D = O\left(\frac{t \log t}{\gamma^2}\right).$$

Substituting this into the sample complexity bound from Theorem E.2, we get:

$$m(\epsilon, \delta) = O\left(\frac{1}{\epsilon^2}\left(\frac{t \log t}{\gamma^2} \log^2\left(\frac{1}{\epsilon}\right) + \log\left(\frac{1}{\delta}\right)\right)\right).$$

Thus, the sample complexity of learning a $\gamma$-fat polytope is as stated. $\square$

