# OpenReview forum: "Learning Safety Constraints for Large Language Models"
_ICML.cc/2025/Conference — ICML 2025 spotlightposter_

### Official Review · Reviewer_ZBtn · 2025-03-06

**Overall Recommendation:** 3

**Summary:**

The study proposes a geometric approach called SaP (Safety Polytope) for large language models (LLMs) to mitigate safety risks. SaP learns and enforces linear safety constraints directly in the model's representation space, identifying safe and unsafe regions. Experiments show it reduces adversarial attack success rates and provides interpretable insights into its safety mechanisms.

**Claims And Evidence:**

The claims are supported by the evidence. For example, SaP operates post-hoc in the representation space without modifying model weights. The approach relies on modifying the internal activations of LLMs rather than retraining or fine-tuning.

**Essential References Not Discussed:**

The MDP modelling is a key concept of the paper. While there is some exploration work on modelling LLMs as MDP is missing:
- Li, Kenneth, Oam Patel, Fernanda Viégas, Hanspeter Pfister, and Martin Wattenberg. "Inference-time intervention: Eliciting truthful answers from a language model." Advances in Neural Information Processing Systems 36 (2023): 41451-41530.
- Song, Da, Xuan Xie, Jiayang Song, Derui Zhu, Yuheng Huang, Felix Juefei-Xu, and Lei Ma. "Luna: A model-based universal analysis framework for large language models." IEEE Transactions on Software Engineering (2024).

**Experimental Designs Or Analyses:**

The paper evaluates SaP’s ability to reduce attack success rates against seven adversarial attack methods, including gradient-based (e.g., GCG, GBDA) and human-crafted jailbreaks.

**Methods And Evaluation Criteria:**

The proposed method, SaP, indeed address the problem of learning safety constraints automatically for LLMs.

**Other Comments Or Suggestions:**

N/A

**Other Strengths And Weaknesses:**

The scalability of the work is doubtable. When the dataset gets very large, I am not sure whether the learned safety constraints are still effective.

**Questions For Authors:**

How scalable is the proposed method?

**Relation To Broader Scientific Literature:**

The paper address the gap of leverage token vectors to build a safety constraint for LLM.

**Theoretical Claims:**

N/A

---

> ### Author Rebuttal · Authors · 2025-03-31
>
> Thank you for your thoughtful review and accurate summary of our paper. We appreciate your recognition that our "claims are supported by the evidence" and that "SaP indeed addresses the problem of learning safety constraints automatically for LLMs."
>
> **Regarding Scalability**
>
> We appreciate your question about the scalability of our approach. We acknowledge that polytope learning presents inherent scalability challenges, which is precisely why we adopted the Convex Polytope Machine (CPM) algorithm instead of traditional methods like QuickHull. This design choice was deliberate to address the computational complexity issues in high-dimensional representation spaces.
>
> Our empirical results demonstrate CPM's effectiveness at scale:
>
> On the BeaverTails dataset (330K examples), our method achieves performance comparable to MLPs with similar parameter counts (Appendix D.3, Table 12).
> For the HarmBench defense task with ~890K data points, SaP achieves an average safety classification accuracy of 91.3% and maintains attack success rates below 5%, further validating its effectiveness at scale.
>
> Based on these results, we expect CPM to perform robustly (on par with MLP) on datasets of similar or even larger scale. In Section 6, we discuss future directions for improving beyond CPM, including references to promising work like Hashimoto et al. (2023) on neural polytope learning.
>
> For practical applications, the inference-time steering mechanism remains efficient regardless of training data size, as it only requires a forward pass through the concept encoder followed by verification against the learned polytope facets.
>
> **Regarding Missing References**
>
> Thank you for suggesting the valuable references (Li et al., 2023 and Song et al., 2024) on MDP modeling for LLMs. We will incorporate these references in our revised manuscript to strengthen the discussion on using MDPs as a theoretical foundation for language model behavior. These papers provide important context for our approach to modeling language generation as a constrained MDP.
>
> We greatly appreciate your feedback and we are happy to address any further questions or comments.
>
> Having addressed your concerns, we would appreciate your feedback during this discussion period. Let us know if there are any further questions that we can clarify, otherwise, we would appreciate it if you would consider increasing your score.

---

### Official Review · Reviewer_aHnQ · 2025-03-09

**Overall Recommendation:** 4

**Summary:**

The authors propose to map unsafe model responses to save model regions in representation space without adjusting the weights of the respective model. Specifically, they represent safety constraints via polytopes and filter responses by assessing the similarity of latent features to the learned polytope. The method restricts models from generating safe outputs while model capabilities are maintained.

References for the remaining review:

[1] Chao et. al., "Jailbreaking Black Box Large Language Models in Twenty Queries", 2023

[2] Andriushchenko et. al., “Jailbreaking Leading Safety-Aligned LLMs with Simple Adaptive Attacks”, 2024

[3] Schwinn et. al., "Soft Prompt Threats: Attacking Safety Alignment and Unlearning in Open-Source LLMs through the Embedding Space", 2024

[4] Carlini et. al., "Adversarial Examples Are Not Easily Detected: Bypassing Ten Detection Methods", 2017

## update after rebuttal

Most of my concerns were addressed. I still believe it's highly relevant (even in the scope of non-adversarial evaluations) to assess the refusal/safety trade-off of this defense. XS-Test, for example, has only ~100 prompts and would give insights into the overrefusal behavior.

I will followed the discussion and increased my score from 2 to 4 and recommend accepting this work.

**Claims And Evidence:**

The authors claim that their proposed algorithm can prevent harmful behavior in LLMs by steering latent activations associated to harmfulness to safe regions in the latent space.

They back up their claims through a large-scale empirical study using 7 different adversarial attacks (excluding recent Sota attacks), three different LLMs, and two common harmfulness benchmarks.

Moreover, they conduct several ablation studies to verify the contribution of individual design choices within their algorithm

**Essential References Not Discussed:**

I am not aware of a paper that I would deem essential to discuss in the context of the presented work that is currently missing from the paper.

**Experimental Designs Or Analyses:**

The results of the conducted experiments are consistent with the results reported in the literature. I did not find anything unusual. However, as described in Methods And Evaluation Criteria some of the experiments are not suitable to back up the claims of the paper.

**Methods And Evaluation Criteria:**

- The benchmarks datasets are widely used in the research field and appropriate
- The used adversarial attacks are mostly weak (GBDA, PEZ , Human Jailbreak, Direct Request). Even GCG may be viewed as weak considering that multiple more efficient GCG variations have been proposed in the literature since the original paper.
- The utility benchmark is not suitable. LLMs can achieve high scores on MMLU without being able to generate coherent sentences.
- Measures for harmfulness evaluations are appropriate

**Other Comments Or Suggestions:**

Text in Figure 4 is not readable. Can the authors increase the font size of the axis labels? Numbers are less relevant. Colorbar could be helpful as well.

**Other Strengths And Weaknesses:**

**Strengths**
- The authors propose a novel method to steer model activations from harmful regions to safe regions
- The authors conduct several ablation studies to investigate the individual components of their mechanism

**Between Strength and Weakness**
- The authors use several adversarial attacks and conduct experiments on multiple models. However, used attacks are not sufficient

**Weakness**
- The authors only use MMLU to analyze model capabilities
- The authors do not try to adaptively break their defense. Detecting adversarial examples (e.g., in latent space) has shown to be very difficult [4]. Adaptive evaluations are necessary to evaluate new defenses.

**Questions For Authors:**

- Could the authors conduct evaluations with stronger and more recent attacks?
- Could the authors conduct a sanity check with very strong continuous attacks if they are able to bypass their defense? A negative result would point towards errors in their evaluation [3]
- Could the authors add more suitable utility evaluation benchmarks to investigate model capability? E.g., MT-Bench and over-refusal benchmarks

# After the rebuttal

Most of my concerns were addressed. I still believe its highly relevant (even in the scope of non-adversarial evaluations), to assess the refusal/safety trade-off of this defense. XS-Test, for example, has only ~100 prompts and would give insights into the overrefusal behavior.

I will follow the discussion and am currently leaning toward increasing my score to three if no other major concerns are raised by the other reviewers.

**Relation To Broader Scientific Literature:**

The authors position their paper appropriately within the related work. However, some references/comparisons with more recent adversarial attacks are missing e.g., [1, 2]

**Theoretical Claims:**

N/A

---

> ### Author Rebuttal · Authors · 2025-03-31
>
> We sincerely thank you for your thoughtful review. We particularly appreciate your recognition of our paper's key strengths:
> - The "novel method to steer model activations from harmful regions to safe regions"
> - Our "several ablation studies to investigate the individual components of [our] mechanism"
> - The comprehensive evaluation across "several adversarial attacks" and "multiple models"
>
> **Clarification on Paper Focus and Contributions**
>
> We would like to first clarify that SaP is not primarily a jailbreak defense paper. Our key contribution is reformulating safety as a geometric constraint learning problem in representation space, which provides both interpretable insights (through facet specialization) and an effective control mechanism (which includes but is not limited to defense). This geometric perspective offers a principled framework for conceptualizing and controlling model safety behaviors. Throughout the paper, we consistently frame our approach in terms of "safety" and "constraints" in their general sense:
> 1. In Section 2, we discuss how "safe and ethical language not only tries to maximize a reward function, but is also subject to some constraints. For instance, humans naturally avoid using language that would hurt someone's feelings, incite harm, or solicit unlawful actions."
> 2. Our extensive experiments on BeaverTails (Sections 4.2-4.3) span 14 distinct safety categories, including animal abuse, discrimination, privacy violations, and misinformation—far beyond just jailbreak prevention.
>
> That said, we agree that stronger defense performance can better support our perspective of geometric constraint learning. We therefore conducted the additional experiments you requested.
>
> **Regarding Stronger and Adaptive Attacks**
> As you suggested, we have evaluated our defense against stronger and more recent attacks. In our response to Reviewer FoeD, we report results on AutoDAN and adaptive attacks. Please refer to it for detailed setting and discussions. Notably:
> 1. For AutoDAN, our method reduces ASR from 66.75% to 1.35% on Ministral-8B, from 1.77% to 0% on Llama2-7B, and from 45% to 0.80% on Qwen2-1.5B.
> 2. For adaptive attacks (on AdvBench), our method achieves 0% ASR for Llama2-7B by directly transferring our trained polytope and hyperparameter settings. With a slight increase in unsafe penalty, Ministral-8B can be steered from 100% ASR to 12%.
>
> **Regarding Utility Evaluation**
> > Reviewer concern: "Could the authors add more suitable utility evaluation benchmarks to investigate model capability? E.g., MT-Bench and over-refusal benchmarks"
>
> We agree that MMLU alone may not fully represent model capabilities. Following your suggestion, we conducted additional evaluations on MT-Bench:
>
> | **Model** | **First Turn Score** | **Second Turn Score** |
> | - | - | - |
> | Ministral-8B | 7.55 | 6.98 |
> | Ministral-8B + SaP | 9.01 | 7.71  |
> | Llama2-7B | 6.54 | 5.46 |
> | Llama2-7B + SaP | 6.98 | 6.21 |
> | Qwen-1.5B | 6.90 | 5.15 |
> | Qwen-1.5B + SaP | 6.85 | 5.06 |
>
> While the results suggest that models with SaP maintain performance and can generate coherent sentences without over-refusing on MT-Bench, we do not intend to claim that SaP improves performance. The slight differences observed might be due to the inherent noise in GPT-based evaluations and the limited sample size (80 evaluation examples). We manually inspected the outputs and did not find any nonsensical or rejection responses. The results are attached in the following URL: https://limewire.com/d/asA5e#Rlx4uqoGNp
>
> Regarding your suggestion to evaluate on benchmarks like OR-Bench, we see this as a valuable future direction. In our preliminary investigations with sensitive but benign prompts, we observe that SaP maintains high performance with minimal false positives. We would like to highlight again that the main contribution of this paper does not directly concern with proposing a new SoTA defense method, but rather to propose a new framework for explicit modeling of safety in LLMs.
>
> **Conclusion**
> We thank you for your constructive feedback. The additional experiments have strengthened our confidence in SaP's learned safety knowledge, which supports our central contribution: reformulating language model safety as a geometric constraint learning problem in representation space. We believe this geometric perspective offers a principled framework for conceptualizing and controlling model safety behaviors beyond just jailbreak defense.
>
> We hope our response addresses the reviewer’s remaining concerns, and given that we could address all the other concerns raised in the review, we would appreciate it if the reviewer would consider increasing our score. We are happy to answer any other open questions. Thanks again for your active engagement!

---

### Official Review · Reviewer_FoeD · 2025-03-13

**Overall Recommendation:** 3

**Summary:**

The paper introduces SaP, a post-hoc safety mechanism that defines a convex polytope in an LLM’s feature space. Using a Concept Encoder to disentangle safety-related features, it learns linear constraints that steer unsafe outputs into a safe region without retraining the model. Experiments show that SaP dramatically reduces adversarial attack success rates while maintaining overall model performance.

**Claims And Evidence:**

The claims are supported by a series of experiments on multiple LLMs, but I am not confident in some of them; please see the section for questions for the authors.

**Essential References Not Discussed:**

all related works have been cited, and [1] from anthropic is also quite related to the topic shown in the paper (though it is released after the submission deadline, it could be added in the future version)

[1] Sharma, M., Tong, M., Mu, J., Wei, J., Kruthoff, J., Goodfriend, S., ... & Perez, E. (2025). Constitutional classifiers: Defending against universal jailbreaks across thousands of hours of red teaming. arXiv preprint arXiv:2501.18837.

**Experimental Designs Or Analyses:**

I have some concerns regarding the experimental design/ evaluation; please see the questions for the authors section.

**Methods And Evaluation Criteria:**

it makes sense.

**Other Comments Or Suggestions:**

the writting looks good

**Other Strengths And Weaknesses:**

Strengths:
1. introduce geometric safety constraints to detect harmful response.
2. post-hoc mechanism preserves model performance while dramatically reducing adversarial attacks.
3. provides interpretable safety facets, offering insights into distinct harmful content categories.

Weaknesses:
1. the LLMs used in the paper are somewhat outdated; Llama 3.1 8B would be preferable to Llama2-7B, but I admit it is just a minor problem.
2. more discussion on the influence of the LLM's size on the proposed method is expected.
3. the paper only considers gradient-based attacks (which optimize for unreadable adversarial strings) and does not consider other attack methods like AutoDan, which optimize for readable adversarial strings.

**Questions For Authors:**

Below is my main concern about this paper. If I have misunderstood anything, please correct me, and I would be happy to raise my score if the authors address these concerns well.

From my perspective, the proposed method first trains the polytope based on the collected data (using only the last hidden representation of the final token). Once the polytope is obtained, during evaluation, it is applied at each token's position; if a token's representation is out of bounds, it is mapped back into the polytope. I then have several questions:

(1) **Overfit Problem**: when training the polytope using gradient-based methods such as GCG, there is a concern that it might overfit to the specific attack pattern rather than learning robust safety constraints. GCG typically generates adversarial suffixes that are unreadable and have high perplexity. As a result, the hidden representations capture these characteristics through self-attention, and the polytope may rely on them for safe/unsafe classification instead of truly learning generalized safety constraints. This raises questions about defense transferability: if the polytope is trained on gradient-based attacks, will it effectively defend against other attack types such as human-crafted jailbreaks, AutoDan [1], or rule-based attacks [2], and vice versa?

(2) **Refusal Pattern After Training**: Building on the previous point, I am surprised that simply mapping the representation back to the trained-internal polytope—as outlined in Algorithm 1—can prevent harmful responses. However, the paper does not provide examples of what the resulting refusal responses look like. Will these responses resemble the traditional refusal pattern, such as “I can’t…”? I am curious about the LLM's response style when the representation is mapped back, and it also raises the question of whether enforcing the safety constraint on only the initial tokens is sufficient to prevent attacks, that will improve the efficiency and reduce false positive rate.

(3) **Unfair Baseline Comparison**: SaP is trained using the top 3 most effective attacks, which are also included in the evaluation. This means that the most effective attack patterns are already "leaked" during training for SaP but not for the other baselines. I recommend that the authors include held-out attacks—such as AutoDan [1] or rule-based attacks [2]—to ensure a fairer comparison.

(4) **Simple MLP may also work**: Following from (3), if one were to directly train a simple MLP for binary classification on the hidden representations using the representation from the same top 3 attacks, I doubt it could achieve a much lower ASR than SaP while maintaining standard MMLU accuracy. The motivation comes from [4].

(5) **MMLU does not prove that standard performance is maintained**: It is not surprising that MMLU performance does not drop because the prompts in MMLU are benign and do not include topics like fraud, hate, or violence as seen in HarmBench. In such cases, the polytope—or even a simple MLP—can easily learn and separate these patterns. In my view, it would be more effective to demonstrate that the false positive rate remains low when the LLM with the polytope is prompted with benign questions involving sensitive topics such as fraud, hate, or violence. This would show whether the polytope mistakenly rejects benign responses involving sensitive topics and prove that it is enforcing constraints based on the harmfulness of the response rather than merely the topic.

[1] Liu, X., Xu, N., Chen, M., & Xiao, C. AutoDAN: Generating Stealthy Jailbreak Prompts on Aligned Large Language Models. In The Twelfth International Conference on Learning Representations.

[2] Andriushchenko, M., Croce, F., & Flammarion, N. (2024). Jailbreaking leading safety-aligned llms with simple adaptive attacks. arXiv preprint arXiv:2404.02151.

[3] Zou, A., Phan, L., Chen, S., Campbell, J., Guo, P., Ren, R., ... & Hendrycks, D. (2023). Representation engineering: A top-down approach to ai transparency. arXiv preprint arXiv:2310.01405.

[4] Sharma, M., Tong, M., Mu, J., Wei, J., Kruthoff, J., Goodfriend, S., ... & Perez, E. (2025). Constitutional classifiers: Defending against universal jailbreaks across thousands of hours of red teaming. arXiv preprint arXiv:2501.18837.

**Relation To Broader Scientific Literature:**

The paper builds on work in constrained decision processes and safe RL by framing safety as linear constraints in LLM representations. It leverages ideas from convex optimization and interpretability research to propose a novel post-hoc safety mechanism.

**Theoretical Claims:**

I have checked the math in the main paper, and it looks reasonable to me.

---

> ### Author Rebuttal · Authors · 2025-03-31
>
> We sincerely thank the reviewer for their thoughtful analysis and valuable feedback.
>
> We would like to first clarify that SaP is not primarily a jailbreak defense paper. Our key contribution is reformulating safety as a geometric constraint learning problem in representation space, which provides both interpretable insights and an effective control mechanisms. Throughout the paper, we consistently frame our approach in terms of "safety" and "constraints" in their general sense. Please see our reply to Reviewer aHnQ for an expanded explanation on this.
>
> **Specific Responses to Concerns**
> > "Overfit Problem"
>
> We thank you for this insightful comment. We agree that only training on GCG will cause SaP to overfit to its specific attack pattern. However, this is not directly a weakness of SaP, as it can be easily mitigated by training it with data collected from diverse sources. This is to be expected, as machine learning models, SaP being no exception, are generally only as good as their training data (no free lunch).
>
> To address the concern about overfitting to specific attack patterns, we conducted additional experiments with attacks not seen during training. The polytopes used in the following experiments are the same ones we trained for reporting the main results in the paper, with no access to any new information.
>
> AutoDAN ASR (5 seeds):
>
> | **Method** | **Ministral-8B** | **LLaMA2-7B** | **Qwen2-1.5B** |
> | - | - | - | - |
> | Original | 66.75  | 1.77 | 45 |
> | Original + SaP | 1.35 ± 0.49 | 0.00 ± 0.00 | 0.80 ± 0.82 |
>
> Adaptive attack:
>
> | **Model Configuration**       | **Attack Success Rate (%)** |
> | - | - |
> | Llama2 Original | 100 |
> | Llama2 + SaP | 0 |
> | Ministral Original | 100 |
> | Ministral + SaP | 98 |
> | Ministral + SaP (increased λ) | 12 |
> | Qwen2 Original | 100 |
> | Qwen2 + SaP | 88 |
>
> The adaptive attack experiments were conducted on AdvBench, following the original released code implementation, which contains requests not seen in HarmBench.
> To the best of our knowledge, three papers to date report defense performance on adaptive attacks, and the most comparable result is from Yi et al., 2025, where they also train their defense methods on HarmBench and test on AdvBench for adaptive attacks. Their best reported performance across 9 defense methods on Llama3-8B is 61% ASR (Table 2 of Yi et al.), while our method achieves 0% ASR for Llama2-7B.
> For Ministral, we found that a direct transfer of SaP hyperparameters trained on HarmBench doesn't defend well against adaptive attacks (98% ASR), but increasing $\lambda$ (the unsafe penalty) can substantially reduce this to 12% ASR. For Qwen2, we discovered that adaptive attack learns to induce harmful responses in Chinese, which our English-trained SaP can't effectively counter. This highlights an interesting direction for future work on multi-lingual adversarial defense through geometric constraints.
>
> > "Refusal Pattern After Training"
>
> Throughout numerous adversarial inputs, the polytope learns to steer models toward safety in a nuanced way. We observe that it guides models to incorporate common refusal phrases like "Sorry", "can't answer this question", "I'm just an AI", etc. We are happy to include qualitative examples in the revised manuscript, if requested.
>
> > "Unfair Baseline Comparison"
>
> We appreciate this concern. Our additional experiments with AutoDAN and adaptive attacks address this by testing on both attack types and adversarial requests not seen during training (generalization from HarmBench to AdvBench). These results strongly support the generalization capabilities of SaP beyond specific patterns used in training.
>
> > "Simple MLP may also work"
>
> From our experiments on BeaverTails, we do see that SaP performs similarly to an MLP with comparable parameter count. However, rejecting based on binary classification might result in over-rejection, which we experimented with and reported in Figure 2, Rejection Sampling (RJ). It reduces Ministral's MMLU performance from 63.4% to 28.5%. Furthermore, polytope constraints offer interpretability benefits that are not possible with a standard MLP, as discussed in our paper's Section 4.2.
>
> > "MMLU does not prove that standard performance is maintained"
>
> Regarding your suggestion to evaluate on benchmarks like OR-Bench, we see this as a valuable future direction. In our preliminary investigations with sensitive but benign prompts, we observe that SaP maintains high performance with minimal false positives. We would like to highlight again that the main contribution of this paper does not directly concern proposing a new SoTA defense method, but rather to propose a new framework for explicit modeling of safety in LLMs.
>
> We thank you once again for your feedback and hope our responses adequately address your concerns. We'd be happy if you would consider revising our score, and we are happy to answer any other open questions. Thanks again for your feedback and active engagement in the review process.

---

> > ### Comment · Reviewer_FoeD · 2025-04-04
> >
> > Thank you for the detailed response. Most of my concerns have been addressed; however, I have a few follow-up questions regarding the “Simple MLP may also work” part:
> >
> > 1. **Scalability:** You mentioned that SaP performs similarly to an MLP when both have comparable parameter counts. Considering that MLPs can be scaled up relatively easily—by using additional layers and larger hidden dimensions (even in this case, the total number of the parameters for it will still not be huge as it's just an MLP) —how would a larger MLP perform relative to SaP? Additionally, can SaP also be scaled up effectively to provide further performance improvements?
> >
> > 2. **Over-Rejection in Binary Classification:** The response noted that rejecting based on binary classification can lead to over-rejection, yet the overall performance between the MLP and SaP remains similar. This seems somewhat contradictory. Besides, MLP can still have the flexibility to implement a workflow similar to that described in Algorithm 1:
> >
> >    For instance, if MLP($\bar{\pi}_l(\boldsymbol{x})$) predicts that the input is safe, we could simply use $\bar{\pi}_l(\boldsymbol{x})$; if it predicts unsafe, we might employ a binary search between 0 and $\bar{\pi}_l(\boldsymbol{x})$ to identify a point $h$—closest to $\bar{\pi}_l(\boldsymbol{x})$—where MLP($h$) predicts safe (given that MLP(0) is definitely safe, such a point must exist). In this way, it performs similarly with SaP, and there will be no over-rejection.
> >
> >    This is just a simple strategy, and there are more complex ways to achieve it, as the problem here is analogous to finding the decision boundary of an MLP's prediction—a topic that has been widely explored in previous literature, especially in the context of boundary-based attacks. But I agree that in this case, the Interpretability may be lost, I am just curious about the performance.
> >
> > 3. **Interpretability vs. Complexity:** You highlighted that the polytope constraints in SaP offer interpretability benefits that a standard MLP cannot provide. However, these constraints might also limit the model’s capacity, potentially resulting in reduced performance. That’s why I brought up the MLP—to understand how much performance is sacrificed in SaP in exchange for interpretability. In other words, when using a standard, unconstrained MLP, what is the performance gap compared to SaP?
> >
> > I appreciate any insights you can provide on these points!

---

> > > ### Author Response · Authors · 2025-04-07
> > >
> > > Thank you for your thoughtful questions about SaP compared to MLPs.
> > >
> > > References:
> > >
> > > [1] Gao, Leo, Tom Dupré la Tour, Henk Tillman, Gabriel Goh, Rajan Troll, Alec Radford, Ilya Sutskever, Jan  Leike, and Jeffrey Wu. "Scaling and evaluating sparse autoencoders." (2024).
> > >
> > > [2] Kantchelian, Alex, Michael C. Tschantz, Ling Huang, Peter L. Bartlett,  Anthony D. Joseph, and J. Doug Tygar. "Large-margin convex polytope machine." Advances in Neural Information Processing Systems 27 (2014).
> > >
> > > **On Scalability:**
> > > > [...] scaled up for performance improvements
> > >
> > > For scaling without preserving interpretability, one could simply add more layers to the concept encoder. One can see the model as being the same as an MLP until the second last layer, and change the MLP's last layer from linear to a polytope. In our experiments, adding one layer (the concept encoder) significantly improves the polytope performance compared to versions without it.
> > >
> > > For scaling while maintaining interpretability, besides adding more layers, one might also need to scale the sparse autoencoder (SAE) architecture. This is possible from the line of SAE scaling research such as [1], which explores designs of SAE that can be effectively scaled to handle complex models while preserving their interpretability advantages.
> > >
> > > We believe scaling could improve interpretability while maintaining safety and capability. Since our work proposes a new framework for safety in LLMs, our focus is to demonstrate that, under the simplest modifications, one could already balance safety and capability, and we welcome future research on scaling it up for more complex problems.
> > >
> > > **On Over-Rejection:**
> > > > [...] lead to over-rejection, yet the overall performance [...] remains similar.
> > >
> > > When we noted similar performance between MLP and SaP, we were referring to classification accuracy. Unlike binary classification that leads to over-rejection by treating all unsafe predictions equally, SaP takes into account the magnitude of constraint violations during steering. This allows for minimal modifications to outputs that might be just slightly over the safety threshold, rather than rejecting them entirely. This difference is evident in Figure 2, comparing rejection sampling with SaP on MMLU.
> > >
> > > > Implement a workflow similar to Algorithm 1
> > >
> > > The workflow you described is precisely our experiments in Figure 6, where we compare 1-facet polytopes (i.e., MLP) with polytopes with more facets. MLP generally underperforms multi-facet versions. This aligns with classic results comparing max-margin SVMs (single decision boundary) with polytopes (multiple boundaries). By [2], under the same feature space, polytope provides a greater modeling capacity and can create larger decision margins. Leveraging the polytope's large-margin nature is an interesting future research direction in studying against bounded attacks. For more detailed discussions on max-margin SVM vs. polytopes, please refer to [2]. Regarding MLP performance, please see the results below.
> > >
> > > **On Interpretability vs. Complexity:**
> > > > Performance gap compared to MLP?
> > >
> > > As we discussed, we do not observe performance degradation in SaP. On BeaverTails, it performs on par with an MLP with similar parameter count. To address your question about a standard MLP, we experimented with these steps:
> > >
> > > (1) Train an MLP based on BCE loss with safety labels.
> > > (2) Use the same algorithm as our steering method (Algorithm 1), replacing the polytope with MLP.
> > >
> > > We experimented with AutoDAN and adaptive attack. The MLP we experimented with has a comparable parameter count to SaP, and we additionally experimented with a version where we added 5 extra layers.
> > >
> > > AutoDAN ASR:
> > > | **Method** | **Ministral-8B** | **LLaMA2-7B** | **Qwen2-1.5B** |
> > > | - | - | - | - |
> > > | Original | 66.75 | 1.77 | 45 |
> > > | SaP | 1.35 ± 0.49 | 0.00 ± 0.00 | 0.80 ± 0.82 |
> > > | MLP | 20 | 0.25 | 1.5 |
> > > | MLP + 5 layers | 20 | 0.25 | 1.25 |
> > >
> > > Adaptive Attack ASR:
> > > | **Method** | **Ministral-8B** | **LLaMA2-7B** | **Qwen2-1.5B** |
> > > | - | - | - | - |
> > > | Original | 100 | 100 | 100 |
> > > | SaP | 12 | 0 | 88 |
> > > | MLP | 100 | 46.81 | 100 |
> > > | MLP + 5 layers | 100 | 0 | 100 |
> > >
> > > On AutoDAN, both MLP versions can perform reasonably on Llama2-7B and Qwen2-1.5B (though worse than SaP), but their Ministral performance is far worse than SaP. For adaptive attacks, while deeper MLPs can match SaP performance on LLaMA2-7B, they fail on Ministral and Qwen where SaP maintains some robustness. This suggests SaP's geometric modeling offers advantages that cannot be easily replicated by simply scaling MLPs by a few layers.
> > >
> > > These experiments reinforce our paper's primary contribution of introducing a geometric perspective that naturally disentangles safety concepts while offering effective model control mechanisms (which are not limited to defense). Thank you for raising these questions. We believe we have addressed your concerns and would appreciate it if you could consider raising your score based on our responses.

---

### Official Review · Reviewer_BjPZ · 2025-03-13

**Overall Recommendation:** 3

**Summary:**

The paper presents a novel approach to increase safety and the adversarial robustness of LLM. Instead of fine-tuning the parameter of the model for safety alignment, the introduced approach SaP (Safety Polytope) is applied during inference by enforcing linear safety constraints using Convex Polytope Machines in the model's representation space or—inspired by sparse autoencoders—a projection (the concept encoder) of the model representation to a higher but sparse dimensional space. By steering the activation subsequently of detecting the representation of a unsafe concept SaP influence the next token prediction during the sampling process.

**Claims And Evidence:**

The paper's primary claim is that SaP provides an inference-time safety mechanism for LLMs that increases safety while maintaining model accuracy.

This is supported by:

1) Empirical evidence on the harmbench dataset while applying adversarial attacks to elicit unsafe behaviour and simultanously evaluating maintaining general LLM performance using MMLU. Considering both SaP outperforms the selected baselines.
2) Further, the paper provides an interpretability analysis demonstrating that different safety constraints specialize in detecting specific types of unsafe content.

However, the empirical evaluation lacks a comparison to related inference time steering methods such as 1) steering vectors (see [1] and [2] or see [3] for an overview of other related detection/steering approaches) and 2) especially SAE which the presented approach even draws inspirations from.

**References**

[1] Wang et al. Model Surgery: Modulating LLM's Behavior Via Simple Parameter Editing. (2024):  https://arxiv.org/abs/2407.08770v1

[2] Lee et al. A Mechanistic Understanding of Alignment Algorithms: A Case Study on DPO and Toxicity (ICML 2024): https://arxiv.org/abs/2401.01967

[3] https://arxiv.org/pdf/2501.17148 While the benchmark was probably released after the ICML submission deadline, the paper provides an overview of (safety) detection/steering algorithms related to the presented method.

**Essential References Not Discussed:**

See above.

**Experimental Designs Or Analyses:**

In general, the experiments are well-structured, though additional comparisons with related safety methods would strengthen the evaluation.

Specifically, the experimental section evaluates SaP on three LLMs, namely Llama2-7B, Ministral-8B, and Qwen2-1.5B.
Additionally, ablations on the impact of the concept encoder and number of safety constraints are conducted.

**Methods And Evaluation Criteria:**

Yes, the choice of evaluation datasets makes sense for the type of assessment. The authors evaluate the introduced approach on the harmbench dataset while applying a range of adversarial attacks to elicit unsafe behavior. Additionally, the MMLU benchmark is used to demonstrate that the general performance of the model is maintained. Further, the introduced approach is compared against five rejection baselines to demonstrate the advantages of SaP. However, as described above, more related inference-time approaches, such as steering vectors or SAEs, should be considered.

**Other Comments Or Suggestions:**

I suggest considering an additional comparison to the above mentioned safety methods.

**Other Strengths And Weaknesses:**

**Other Strengths**
- Limitations and future work are well discussed.

**Other Weaknesses**
- While the authors state that "our approach scales efficiently to large batches via existing tools for vectorized computation“ the computational overhead during inference is unclear. Therefore some uncertainty of the approach’s practability remains. A additional analysis of the computational overhead during inference would strength the paper.

**Questions For Authors:**

- See comment in supplementary material section:
Could you clarify the number of seeded runs of the different experiments and why you chose a different number of runs for the baselines?

- Could you provide an overview of the additional computational overhead during inference?

**Relation To Broader Scientific Literature:**

The paper is well-grounded in prior literature on LLM safety, and specifically, Section 5 describes the relation to prior literature quite well. However, even if inspired by SAEs, a comparison is missing. Further, the paper lacks a discussion on the relation to other recent inference-time steering methods, as mentioned above.

**Theoretical Claims:**

The paper defines LLM safety as a CMDP problem, where constraints define safe and unsafe regions in the representation space. The claims are supported by references to prior work.

---

> ### Author Rebuttal · Authors · 2025-03-31
>
> We sincerely thank the reviewer for their thoughtful analysis and constructive feedback. We appreciate your recognition of our paper's key contributions:
>
> 1. The novel geometric approach to LLM safety through representation space constraints.
>
> 2. The effectiveness of SaP in defending against adversarial attacks while maintaining model capabilities
>
> 3. The interpretability insights provided by our analysis showing how different safety constraints specialize in detecting specific types of unsafe content.
>
> Below, we address your specific questions and suggestions:
>
> **Regarding experimental methodology:**
>
> > Question: "Could you clarify the number of seeded runs of the different experiments?"
>
> We apologize for the confusion. This is indeed a typo in Appendix C.2. All methods (including baselines) were evaluated over 5 seeds. As shown in Tables 1-4 in the appendix, we report means and standard deviations calculated over 5 seeds for all methods. We will correct this inconsistency in the revised manuscript.
>
> **Regarding computational overhead during inference:**
>
> > Question: "While the authors state that 'our approach scales efficiently to large batches via existing tools for vectorized computation' the computational overhead during inference is unclear. Therefore some uncertainty of the approach's practicability remains."
>
> Thank you for raising this important point. We have conducted additional experiments to quantify the inference cost of SaP. First, we measured the average per-token processing time with and without SaP across all three model architectures:
>
> | **Model**    | **Configuration** | **Avg Time per Token (s)** | **Overhead** |
> | ------------ | ----------------- | -------------------------- | ------------ |
> | Llama2-7B    | with SaP          | 0.0301                     | +29%         |
> | Llama2-7B    | without SaP       | 0.0234                     | -            |
> | Ministral-8B | with SaP          | 0.0381                     | +8%          |
> | Ministral-8B | without SaP       | 0.0353                     | -            |
> | Qwen-1.5B    | with SaP          | 0.0309                     | +35%         |
> | Qwen-1.5B    | without SaP       | 0.0228                     | -            |
>
> While SaP adds approximately 8-35% overhead at the per-token level, this efficiency advantage becomes even more significant when comparing end-to-end runtime for practical tasks. For the MMLU benchmark, all runs conducted on A100 40GB GPUs via clean slurm jobs with no other program interference. For these baselines, we use the implementation from the llm-jailbreaking-defense [benchmark](https://github.com/YihanWang617/llm-jailbreaking-defense).
>
> | **Method**                 | **Total Runtime** | **Relative to Baseline** |
> | -------------------------- | ----------------- | ------------------------ |
> | Llama2-7B (baseline)       | 24 min            | 1.0×                     |
> | Llama2-7B + SaP            | 38 min            | 1.6×                     |
> | Llama2-7B + ICL            | 11h               | 27.5×                    |
> | Llama2-7B + Response Check | 11h 17min         | 28.1×                    |
> | Llama2-7B + Self Reminder  | 10h 42min         | 26.7×                    |
> | Llama2-7B + SmoothLLM      | 11h 35min         | 28.8×                    |
>
> These results demonstrate that SaP provides a dramatically more efficient safety mechanism compared to prompt-based alternatives, requiring only 1.6× the baseline runtime while other methods require 26-29× more time. This substantial efficiency advantage stems from SaP's direct manipulation of model representations rather than relying on multiple forward passes or additional inference steps that characterize most prompt-based approaches.
>
> **Regarding additional comparisons with related methods:**
>
> > Concern: "[...] lacks a comparison to related inference time steering methods such as 1) steering vectors and 2) especially SAEs."
>
> We appreciate this valuable suggestion. Our concept encoder indeed shares implementation similarities with Sparse Autoencoders (SAEs), and we view the polytope constraint mechanism as offering a complementary geometric perspective compared to steering vectors.
>
> We agree these comparisons would strengthen our paper and plan to:
>
> 1. Provide a more thorough analysis of how our concept encoder relates to and differs from SAEs and steering vectors
> 2. Incorporate the suggested references and Wu et al. (2025) in our revised manuscript
> 3. Expand our discussion of related inference-time approaches to better position our work within this literature
>
> We believe these additions will address your concerns and enhance the paper's contribution to the field of LLM safety. Having addressed your concerns, we would appreciate your feedback during this discussion period. Let us know if there are any further questions that we can clarify, otherwise, we would appreciate it if you would consider increasing your score.

---

### Decision · Program_Chairs · 2025-05-01

**Decision:**

Accept (spotlight poster)

**Comment:**

Strong Accept because (1) the paper addresses an important area, namely safety for LLMs, (2) develops a novel geometric approach to enforce safety constraints with interpretable insights, (3) jailbreak defence experiments are convincing and (4) unanimously received accept or weak accept from all reviewers.